# ADVERSARIAL POLICIES BEAT SUPERHUMAN GO AIS

## ABSTRACT

We attack the state-of-the-art Go-playing AI system, KataGo, by training adversarial policies that play against frozen KataGo victims. Our attack achieves a >99% win-rate when KataGo uses no tree-search, and a >77% win-rate when KataGo uses enough search to be superhuman. Notably, our adversaries do not win by learning to play Go better than KataGo—in fact, our adversaries are easily beaten by human amateurs. Instead, our adversaries win by tricking KataGo into making serious blunders. Our results demonstrate that even superhuman AI systems may harbor surprising failure modes. Example games are available at https://go-attack-iclr.netlify.app/.

## 1 INTRODUCTION

Reinforcement learning from self-play has achieved superhuman performance in a range of games including Go (Silver et al., 2016), chess and shogi (Silver et al., 2016), and Dota (OpenAI et al., 2019). Moreover, idealized versions of self-play provably converge to Nash equilibria (Brown, 1951; Heinrich et al., 2015). Although realistic versions of self-play may not always converge, the strong empirical performance of self-play seems to suggest this is rarely an issue in practice.

Nonetheless, prior work has found that seemingly highly capable continuous control policies trained via self-play can be exploited by *adversarial policies* (Gleave et al., 2020; Wu et al., 2021). This suggests that self-play may not be as robust as previously thought. However, although the victim agents are state-of-the-art for continuous control, they are still well below *human* performance. This raises the question: are adversarial policies a vulnerability of self-play policies *in general*, or simply an artifact of insufficiently capable policies?

To answer this, we study a domain where self-play has achieved very strong performance: Go. Specifically, we train adversarial policies end-to-end to attack KataGo (Wu, 2019), the strongest publicly available Go-playing AI system. Using less than 5% of the compute used to train KataGo, we obtain adversarial policies that win >99% of the time against KataGo with no search, and >77% against KataGo with enough search to be superhuman.

Critically, our adversaries do *not* win by learning a generally capable Go policy. Instead, the adversaries trick KataGo into making serious blunders that result in KataGo losing the game (Figure 1). Despite being able to beat KataGo, our adversarial policies lose against even amateur Go players (see Appendix F.1). So KataGo is in this instance less *robust* than human amateurs, despite having superhuman *capabilities*. This is a striking example of non-transitivity, illustrated in Figure 2.

Our adversaries have no special powers: they can only place stones, or pass, like a regular player. We do, however, give our adversaries access to the victim network they are attacking. In particular, we train our adversaries using an AlphaZero-style training process (Silver et al., 2018), similar to that of KataGo. The key differences are that we collect games with the adversary playing the victim, and that we use the victim network to select victim moves during the *adversary's* tree search.

KataGo is the strongest publicly available Go AI system at the time of writing. With search, KataGo is strongly superhuman, winning (Wu, 2019, Section 5.1) against ELF OpenGo (Tian et al., 2019) and Leela Zero (Pascutto, 2019) that are themselves superhuman. In Appendix D, we estimate that KataGo without search plays at the level of a top 100 European player, and that KataGo with 2048 visits per move of search is much stronger than any human.

Our paper makes three contributions. First, we propose a novel attack method, hybridizing the attack of Gleave et al. (2020) and AlphaZero-style training (Silver et al., 2018). Second, we demon-

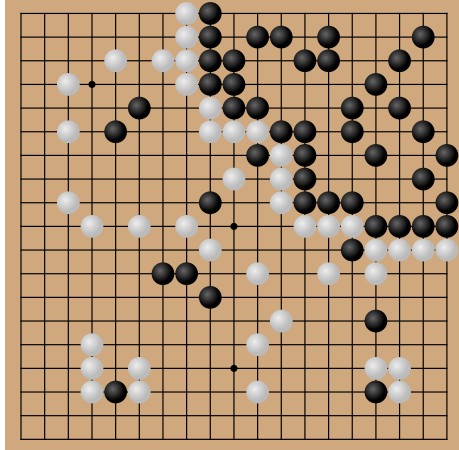
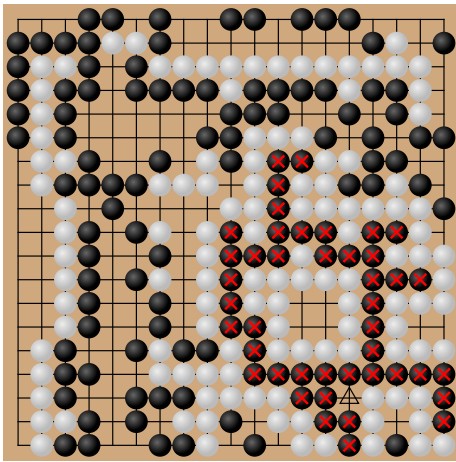

(a) Adversary wins as black by tricking the victim (`Latest`, no search) into passing prematurely, ending the game. Explore the game.

(b) Adversary wins as white by capturing a group (X) that the victim (`Latest_def`, 2048 visits) leaves vulnerable. Explore the game.

Figure 1: Two randomly sampled games against the strongest policy network, `Latest`. (a) An adversarial policy beats KataGo by tricking it into passing. The adversary then passes in turn, ending the game with the adversary winning under the Tromp-Taylor ruleset for computer Go (Tromp, 2014) that KataGo was trained and configured to use (see Appendix A). The adversary gets points for its territory in the top-right corner (devoid of victim stones) whereas the victim does not get points for the territory in the bottom-left due to the presence of the adversary's stones. (b) A different adversarial policy beats a superhuman-level victim immunized against the "passing trick". The adversary lures the victim into letting a large group of victim stones (X) get captured by the adversary's next move (△). Appendix F.3 has a more detailed description of this adversary's behavior.

strate the existence of two distinct adversarial policies against the state-of-the-art Go AI system, KataGo. Finally, we provide a detailed empirical investigation into these adversarial policies, including a qualitative analysis of their game play. Our open-source implementation is available at —anonymized—.

## 2 RELATED WORK

Our work is inspired by the presence of adversarial examples in a wide variety of models (Szegedy et al., 2014). Notably, many image classifiers reach or surpass human performance (Ho-Phuoc, 2018; Russakovsky et al., 2015; Shankar et al., 2020; Pham et al., 2021). Yet even these state-of-the-art image classifiers are vulnerable to adversarial examples (Carlini et al., 2019; Ren et al., 2020). This raises the question: could highly capable deep RL policies be similarly vulnerable?

One might hope that the adversarial nature of self-play training would naturally lead to robustness. This strategy works for image classifiers, where adversarial training is an effective if computationally expensive defense (Madry et al., 2018; Ren et al., 2020). This view is further bolstered by the fact that idealized versions of self-play provably converge to a Nash equilibrium, which is unexploitable (Brown, 1951; Heinrich et al., 2015). However, our work finds that in practice even state-of-the-art and professional-level deep RL policies are still vulnerable to exploitation.

It is known that self-play may not converge in non-transitive games (Balduzzi et al., 2019). However, Czarnecki et al. (2020) has argued that real-world games like Go grow increasingly transitive as skill increases. This would imply that while self-play may struggle with non-transitivity early on during training, comparisons involving highly capable policies such as KataGo should be mostly transitive. By contrast, we find a striking non-transitivity: our adversaries exploit KataGo agents that beat human professionals, yet even an amateur Go player can beat our adversaries (Appendix F.1).

Most prior work attacking deep RL has focused on perturbing observations (Huang et al., 2017; Ilahi et al., 2022). Concurrent work by Lan et al. (2022) shows that KataGo with $\leq 50$ visits can

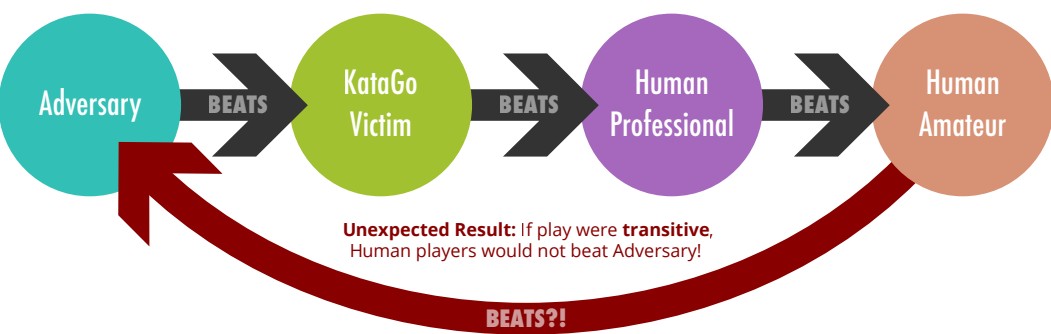

Figure 2: A human amateur beats our adversarial policy (Appendix F.1) that beats KataGo. This non-transitivity shows the adversary is not a generally capable policy, and is just exploiting KataGo.

be induced to play poorly by adding two adversarially chosen moves to the move history input of the KataGo network, even though these moves do not substantially change the win rate estimated by KataGo with 800 visits. However, the perturbed input is highly off-distribution, as the inputs tell KataGo that it *chose* to play a seemingly poor move on the previous turn. Moreover, an attacker that can force the opponent to play a specific move has easier ways to win: it could simply make the opponent resign, or play a maximally bad move. We instead follow the threat model introduced by Gleave et al. (2020) of an adversarial *agent* acting in a shared environment.

Prior work on such *adversarial policies* has focused on attacking subhuman policies in simulated robotics environments (Gleave et al., 2020; Wu et al., 2021). In these environments, the adversary can often win just by causing small changes in the victim's actions. By contrast, our work focuses on exploiting professional-level Go policies that have a discrete action space. Despite the more challenging setting, we find these policies are not only vulnerable to attack, but also fail in surprising ways that are quite different from human-like mistakes.

Adversarial policies give a lower bound on the *exploitability* of an agent: how much expected utility a best-response policy achieves above the minimax value of the game. Exactly computing a policy's exploitability is feasible in some low-dimensional games (Johanson et al., 2011), but not in larger games such as Go which has approximately $10^{172}$ possible states (Allis, 1994, Section 6.3.12). Prior work has lower bounded the exploitability in some poker variants using search (Lisý & Bowling, 2017), but the method relies on domain-specific heuristics that are not applicable to Go.

In concurrent work Timbers et al. (2022) developed the *approximate best response* (ABR) method to estimating exploitability. Whereas we exploit the open-source KataGo agent, they exploit a proprietary replica of AlphaZero from Schmid et al. (2021). They obtain a 90% win rate against no-search AlphaZero and 65% with 800 visits (Timbers et al., 2022, Figure 3). In Appendix D.3 we estimate that their AlphaZero victim with 800 visits plays at least at the level of a top-200 professional, and may be superhuman. That we were both able to exploit unrelated codebases confirms the vulnerability is in AlphaZero-style training as a whole, not merely an implementation bug.

Our attack methodology is similar to Timbers et al.: we both use an AlphaZero-style training procedure that is adapted to use the *opponent's* policy during search. However, unlike Timbers et al. we attempt to model the victim's search process inside our adversary via A-MCTS-R (see Section 4). Additionally, our curriculum uses checkpoints as well as search. Finally, we provide a detailed empirical investigation into how our attack works, including examples of games played by the adversaries, the victim's predicted win rate, the degree to which the adversaries transfer to different victims and an investigation of possible defenses.

## 3 BACKGROUND

### 3.1 THREAT MODEL

Following Gleave et al. (2020), we consider the setting of a two-player zero-sum Markov game (Shapley, 1953). Our threat model assumes the attacker plays as one of the agents, which

we will call the *adversary*, and seeks to win against some *victim* agent. Critically, the attacker does not have any special powers—they can only take the same actions available to a regular player.

The key capability we grant to the attacker is gray-box access to the victim agent. That is, the attacker can evaluate the victim's neural network on arbitrary inputs. However, the attacker does not have direct access to the network weights. We furthermore assume the victim agent follows a fixed policy, corresponding to the common case of a pre-trained model deployed with static weights. Gray-box access to a fixed victim naturally arises whenever the attacker can run a copy of the victim agent, such as a commercially available or open-source Go AI system.

This is a challenging setting even with gray-box access. Although finding an exact Nash equilibrium in a game as complex as Go is intractable, a priori it seems plausible that a professional-level Go system might have reached a *near-Nash* or $\epsilon$-*equilibrium*. In this case, the victim could only be exploited by an $\epsilon$ margin. Moreover, even if there *exists* a policy that can exploit the victim, it might be computationally expensive to find given that self-play training did not discover the vulnerability.

Consequently, our two primary success metrics are the *win rate* of the adversarial policy against the victim and the adversary's *training time*. We also track the mean score difference between the adversary and victim, but this is not explicitly optimized for by the attack. Crucially, tracking training time rules out the degenerate "attack" of simply training KataGo for longer than the victim.

In principle, it is possible that a more sample-efficient training regime could produce a stronger agent than KataGo in a fraction of the training time. While this might be an important advance in computer Go, we would hesitate to classify it as an attack. Rather, we are looking for the adversarial policy to demonstrate *non-transitivity*, as this suggests the adversary is winning by exploiting a specific weakness in the opponent. That is, as depicted in Figure 2, the adversary beats the victim, the victim beats some baseline opponent, and that baseline opponent can in turn beat the adversary.

## 3.2 KATAGO

We chose to attack KataGo as it is the strongest publicly available Go AI system. KataGo won against ELF OpenGo (Tian et al., 2019) and Leela Zero (Pascutto, 2019) after training for only 513 V100 GPU days (Wu, 2019, section 5.1). ELF OpenGo is itself superhuman, having won all 20 games played against four top-30 professional players. The latest networks of KataGo are even stronger than the original, having been trained for over 10,000 V100-equivalent GPU days. Indeed, even the policy network with *no search* is competitive with top professionals (see Appendix D.1).

KataGo learns via self-play, using an AlphaZero-style training procedure (Silver et al., 2018). The agent contains a neural network with a *policy head*, outputting a probability distribution over the next move, and a *value head*, estimating the win rate from the current state. It then conducts Monte-Carlo Tree Search (MCTS) using these heads to select self-play moves, described in Appendix B.1. KataGo trains the policy head to predict the outcome of this tree search, a policy improvement operator, and trains the value head to predict whether the agent wins the self-play game.

In contrast to AlphaZero, KataGo also has a number of additional heads predicting auxiliary targets such as the opponent's move on the following turn and which player "owns" a square on the board. These heads' output are not used for actual game play—they serve only to speed up training via the addition of auxiliary losses. KataGo additionally introduces architectural improvements such as global pooling, and improvements to the training process such as playout cap randomization.

These modifications to KataGo improve its sample and compute efficiency by several orders of magnitude relative to prior work such as ELF OpenGo. For this reason, we choose to build our attack on top of KataGo, although in principle the same attack could be implemented on top of any AlphaZero-style training pipeline. We describe our extensions to KataGo in the following section.

## 4 ATTACK METHODOLOGY

Prior works, such as KataGo and AlphaZero, train on self-play games where the agent plays many games against itself. We instead train on games between our adversary and a fixed victim agent, and only train the adversary on data from the turns where it is the adversary's move, since we wish to train the adversary to exploit the victim, not mimic it. We dub this procedure *victim-play*.

In regular self-play, the agent models its opponent's moves by sampling from its own policy network. This makes sense in self-play, as the policy *is* playing itself. But in victim-play, it would be a mistake to model the victim as playing from the *adversary's* policy network. We introduce two distinct families of *Adversarial MCTS* (A-MCTS) to address this problem. See Appendix C for the hyperparameter settings we used in experiments.

**Adversarial MCTS: Sample (A-MCTS-S)**. In A-MCTS-S (Appendix B.2), we modify the adversary's search procedure to sample from the victim's policy network at *victim-nodes* in the Monte Carlo tree when it is the victim's move, and from the adversary's network at *adversary-nodes* where it is the adversary's turn. We also disable some optimizations added in KataGo, such as adding noise to the policy prior at the root. Finally, we introduce a variant A-MCTS-S++ that averages the victim policy network's predictions over board symmetries, to match the default behavior of KataGo.

**Adversarial MCTS: Recursive (A-MCTS-R)**. A-MCTS-S systematically underestimates the strength of victims that use search since it models the victim as sampling directly from the policy head. To resolve this, A-MCTS-R runs MCTS for the victim at each victim node in the A-MCTS-R tree. Unfortunately, this change increases the computational complexity of both adversary training and inference by a factor equal to the victim search budget. We include A-MCTS-R primarily as an upper bound to establish how much benefit can be gained by resolving this misspecification.

**Initialization**. We randomly initialize the adversary's network. Note that we cannot initialize the adversary's weights to those of the victim as our threat model does not allow white-box access. Additionally, a random initialization encourages exploration to find weaknesses in the victim, rather than simply a stronger Go player. However, a randomly initialized network will almost always lose against a highly capable network, leading to a challenging initial learning problem. KataGo use of auxiliary targets partially alleviates this problem, as the adversary's network can learn something useful about the game even from lost matches.

**Curriculum**. To help overcome the challenging random initialization, we introduce a curriculum that trains against successively stronger versions of the victim. We switch to a more challenging victim agent once the adversary's win rate exceeds a certain threshold. We modulate victim strength in two ways. First, we train against successively later checkpoints of the victim agent, as KataGo releases its entire training history. Second, we gradually increase the amount of search that the victim performs during victim-play.

## 5 EVALUATION

We evaluate our attack method against KataGo (Wu, 2019). In Section 5.1 we find our A-MCTS-S algorithm achieves a 99.9% win rate against `Latest` (the strongest KataGo network) playing without search. Notably `Latest` is very strong even without search: we find in Appendix D.1 that it is comparable to a top 100 European player. We win against `Latest` by tricking it into passing early and losing.

In Section 5.2, we then add a *pass-alive defense* to the victim to defend against the aforementioned attack. The defended victim `Latest`$_{\text{def}}$ is provably unable to lose via accidentally passing and is roughly the same strength as `Latest`.[1] However, we are still able to find an attack that achieves a 99.8% win rate against `Latest`$_{\text{def}}$ playing without search. The attack succeeds against the victim playing with search as well, with our best result in Section 5.3 being a 79.6% win rate against `Latest`$_{\text{def}}$ with 2048 visits.

Our second attack is qualitatively very different from our first attack as it does not use the pass-trick. To check that our defense did not introduce any unintended weaknesses, we evaluate our attack against the unmodified `Latest`. Our second attack achieves a 100% win rate (over 500 games) against `Latest` without search, and a 77.6% win rate against `Latest` with 2048 visits. In Appendix D.2, we estimate that `Latest` with 2048 visits is much stronger than the best human Go players.

---

[1] `Latest`$_{\text{def}}$ won 456 / 1000 games against `Latest` when both agents used no tree-search and 461 / 1000 games when both agents used 2,048 visits / move of search.

## 5.1 ATTACKING THE VICTIM POLICY NETWORK

We train an adversarial policy using A-MCTS-S and a curriculum, as described in Section 4. We start from a checkpoint `cp127` around a quarter of the way through training, until reaching the `Latest` checkpoint corresponding to the strongest KataGo network (see Appendix C.1 for details).

Our best adversarial policy checkpoint playing with 600 visits against `Latest` playing with no search achieves a 99.9% win rate. Notably, this high win rate is achieved despite our adversarial policy being trained for only $3.4 \times 10^7$ time steps—just 0.3% as many time steps as the victim it is exploiting. Critically, this adversarial policy does not win by playing a stronger game of Go than the victim. Instead, it follows a bizarre strategy illustrated in Figure 1 that loses even against human amateurs (see Appendix F.1). The strategy tricks the KataGo policy network into passing prematurely at a point where the adversary has more points.

Appendix E contains further evaluation and analysis of this adversarial attack. Though this adversary was only trained on no-search victims, it transfers to the low-search regime, achieving a 54% win rate against `Latest` playing with 64 visits by increasing the number of visits by our adversarial policy to 4096.

## 5.2 ADDING A DEFENSE TO THE VICTIM POLICY NETWORK

We design a hard-coded defense for the victim against the attack found in Section 5.1: only passing when it cannot change the outcome of the game. Concretely, we only allow the victim to pass when its only legal moves are in its own *pass-alive territory*, a concept described in the official KataGo rules which extends the traditional Go notion of a pass-alive group (Wu, 2021) (see Appendix B.4 for a full description of the algorithm). Given a victim $V$, we let $V_{def}$ denote the victim with this defense applied. The defense completely thwarts the adversary from Section 5.1; the adversary loses every game out of 1600 against `Latest`$_{def}$.

With the defense always enabled for the victim, we again train an adversarial policy using A-MCTS-S. The curriculum starts from an earlier checkpoint `cp39`$_{def}$. After the curriculum reaches `Latest`$_{def}$ playing with no search, the curriculum starts increasing the number of visits used by `Latest`$_{def}$. See Appendix C.2 for curriculum details.

In Figure 3 we evaluate our adversarial policy against the policy networks of `cp39`$_{def}$, `cp127`$_{def}$, and `Latest`$_{def}$. We see that an attack that worked against `Latest`$_{def}$ transfers well to `cp127`$_{def}$ but not to `cp39`$_{def}$, and an attack against `cp39`$_{def}$ early in training did not transfer well to `cp127`$_{def}$ or `Latest`$_{def}$. These results suggest that different checkpoints have unique vulnerabilities.

Our best adversarial policy checkpoint playing with 600 visits against `Latest`$_{def}$ playing with no search achieves a 99.8% win rate. The adversarial policy also still works against `Latest` with the defense disabled, achieving a 100.0% win rate over 500 games. The adversary is trained for $4.98 \times 10^8$ time steps—just 4.2% as many time steps as the `Latest`, but $15\times$ more than the adversary from Section 5.1. Again the adversarial policy loses against human amateurs (see Appendix F.1).

## 5.3 ATTACKING A VICTIM WITH SEARCH

We evaluate the ability of the adversarial policy trained in Section 5.2 to exploit `Latest`$_{def}$ playing *with* search. This adversary was trained on `Latest`$_{def}$ only up to 256 victim visits. Although the adversary using A-MCTS-S with 200 visits achieves a win rate of 100% over 60 games against `Latest`$_{def}$ without search, in Figure 4a we find the win rate drops to 65% at 2048 victim visits and 53% at 4096 victim visits. Although lower, this win rate still demonstrates striking non-transitivity, as a merely average human amateur could beat this adversary (while standing little chance against the victim), and its wins are produced when the victim makes severe mistakes a human would avoid (see Appendix F.3 for further analysis).

This win rate can also be increased by running A-MCTS-S with a greater adversary visit count. In Figure 4b we see that in this case, the win rate against 2048 victim visits seems to plateau beyond 512 adversary visits at 72%–79.6%.

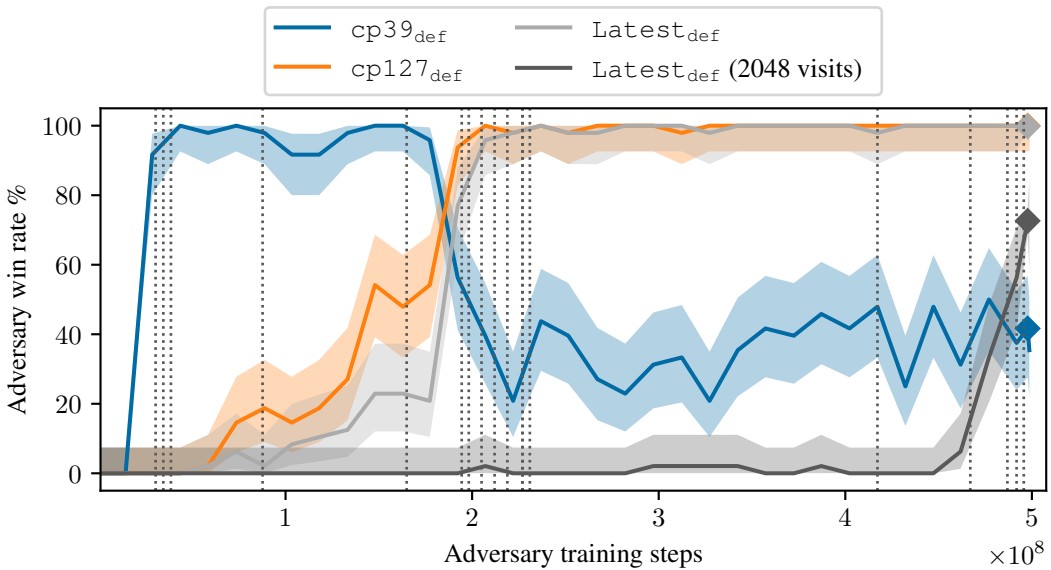

Figure 3: The win rate ($y$-axis) of the adversarial policy over time ($x$-axis) playing with 600 visits against the $cp39_{def}$, $cp127_{def}$, and $Latest_{def}$ victim policy networks playing without search, as well as $Latest_{def}$ playing with 2048 visits. The strongest adversary checkpoint (marked ♦) wins $547/548 = 99.8\%$ games against $Latest_{def}$ without search and $398/548 = 72.6\%$ games against $Latest_{def}$ with 2048 visits. The shaded interval is a 95% Clopper-Pearson interval over 48 games per checkpoint. The adversarial policy is trained with a curriculum, starting from $cp39_{def}$ without search and ending at $Latest_{def}$ with 256 visits. Vertical dashed lines denote switches to a later victim training policy or increasing $Latest_{def}$'s amount of search. See Appendix C.2 for the exact curriculum specification.

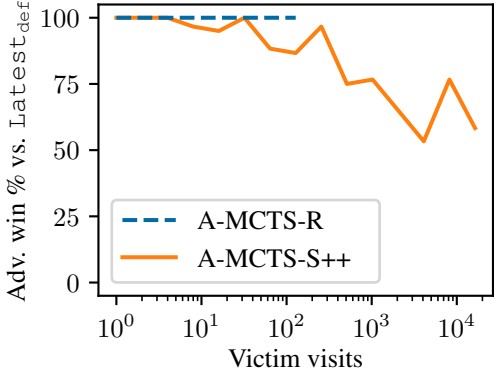

(a) Win rate by number of victim visits ($x$-axis) for A-MCTS-S and A-MCTS-R. The adversary is run with 200 visits.

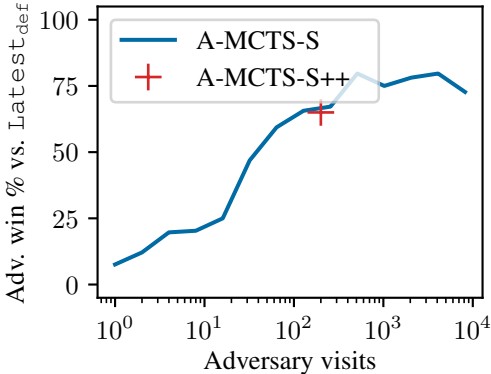

(b) Win rate of A-MCTS-S by number of adversary visits, playing against victim with 2048 visits. Higher adversary visits moderately improve win rates.

Figure 4: We evaluate the ability of the adversarial policy from Section 5.1 to win against the $Latest_{def}$ victim with search.

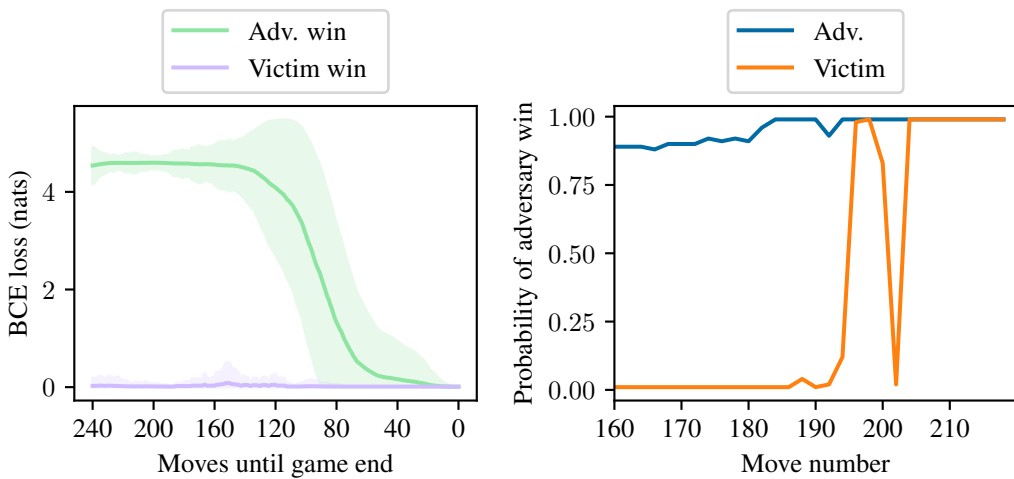

(a) Binary cross entropy loss of the predicted win-rates for `Latest` at 1,600 visits, playing against an adversary at 200 visits. The green curve is averaged over games won by the adversary; the purple curve is averaged over games won by the victim.

(b) Probability of adversary victory according to the adversary and victim value networks, for a portion of a randomly selected game. Note the sudden changes in winrate prediction between moves 196 and 204. Explore the game.

Figure 5: Analysis of `Latest` and adversary's predicted win rate.

A-MCTS-S models the victim as having no search at both training and inference time. Another way to improve performance is to use A-MCTS-R, which correctly models the victim at inference by performing an MCTS search at each victim node in the adversary's tree. A-MCTS-R achieves an impressive 100% win rate against `Latest`$_\mathrm{def}$ up to 128 victim visits over 90 games at each data point. It is too computationally expensive at much higher visit counts, however.

Our adversarial policy with 600 adversary visits achieves a 77.6% win rate versus `Latest` with 2048 visits with the defense disabled, verifying that our policy is not exploiting anomalous behavior introduced by the defense.

## 5.4 UNDERSTANDING THE ATTACK

Qualitatively, the attack we trained in Section 5.2 wins by coaxing the victim into creating a large group of stones in a circular pattern, thereby exploiting a weakness in KataGo's value network which allows it to capture the group. This causes the score to shift decisively and unrecoverably in the adversary's favor.

To better understand the attack, we examined the winrate predictions produced by both the adversary's and the victim's value networks at each turn of a game. As depicted in Figure 5a, the victim's winrate predictions are strikingly uncalibrated in the games where it loses (the majority), with a high cross-entropy loss throughout the bulk of the game. The predictions only improve in accuracy close to the end game.

Typically the victim predicts that it will win with over 99% confidence[2] for most of the game, then suddenly realizes it will lose with high probability, often just *one move* before its circular group is captured. In some games, we observe that the victim's winrate prediction oscillates wildly before finally converging on certainty that it will lose; see Figure 5b for an example. This is in stark contrast to the adversary's own predictions, which change much more slowly and are less confident.

---

[2]Winrate predictions are rounded to the nearest two decimal places before being logged in KataGo.

## 6 LIMITATIONS AND FUTURE WORK

This paper has demonstrated that even agents at the level of top human professionals can be vulnerable to adversarial policies. However, our results do not establish how common such vulnerabilities are: it is possible Go-playing AI systems are unusually vulnerable. A promising direction for future work is to evaluate our attack against strong AI systems in other games.

If professional-level policies can be vulnerable to attack, it is natural to ask how we can *defend* against this possibility. Fortunately, there are a number of promising multi-agent RL techniques. An important direction for future work is to evaluate policies trained with these approaches to determine if they are also exploitable, or if this is an issue limited to self-play.

One promising technique is counterfactual regret minimization (Zinkevich et al., 2007, CFR) that can beat professional human poker players (Brown & Sandholm, 2018). CFR has difficulty scaling to high-dimensional state spaces, but regularization methods (Perolat et al., 2021) can natively scale to games such as Stratego with a game tree $10^{175}$ times larger than Go (Perolat et al., 2022). Alternatively, methods using populations of agents such as policy-space response oracles (Lanctot et al., 2017), AlphaStar's Nash league (Vinyals et al., 2019) or vanilla PBT (Czempin & Gleave, 2022) may be more robust than self-play, although are substantially more computationally expensive.

Finally, we found it harder to exploit agents that use search, with our attacks achieving a lower win rate and requiring more computational resources. An interesting direction for future work is to try and find more effective and compute-efficient methods for attacking agents that use large amounts of search, such as modeling the victim (Appendix B.3).

## 7 CONCLUSION

We trained adversarial policies that are able to exploit a superhuman Go AI. Notably, the adversaries do not win by playing a strong game of Go—in fact, they can be easily beaten by a human amateur. Instead, the adversaries win by exploiting particular blindspots in the victim agent. This result suggests that even highly capable agents can harbor serious vulnerabilities.

The original KataGo paper (Wu, 2019) was published in 2019 and KataGo has since been used by many Go enthusiasts and professional players as a playing partner and analysis engine. However, despite the large amount of attention placed on KataGo, to our knowledge the vulnerabilities discussed in this paper were previously unknown. This suggests that learning-based attacks like the one developed in this paper may be an important tool for uncovering hard-to-spot vulnerabilities in AI systems.

Our results underscore that improvements in capabilities do not always translate into adequate robustness. These failures in Go AI systems are entertaining, but similar failures in safety-critical systems such as automated financial trading or autonomous vehicles could have dire consequences. We believe the ML research community should invest considerable effort into improving robust training and adversarial defense techniques in order to produce models with the high levels of reliability needed for safety-critical systems.

### AUTHOR CONTRIBUTIONS

Removed for double blind submission.

### ACKNOWLEDGMENTS

Removed for double blind submission.

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

## A  RULES OF GO USED FOR EVALUATION

We evaluate all games with Tromp-Taylor rules (Tromp, 2014), after clearing opposite-color stones within pass-alive groups computed by Benson's algorithm (Benson, 1976). KataGo was configured to play using these rules in all our matches against it. Indeed, these rules simply consist of KataGo's version of Tromp-Taylor rules with `SelfPlayOpts` enabled (Wu, 2021). We use a fixed Komi of 6.5.

We chose these *modified Tromp-Taylor* rules because they are simple, and KataGo was trained on (variants) of these rules so should be strongest playing with them. Although the exact rules used were randomized during KataGo's training, modified Tromp-Taylor made up a plurality of the training data. That is, modified Tromp-Taylor is at least as likely as any other configuration seen during training, and is more common than some other options.[3]

In particular, KataGo training randomized between area vs. territory scoring as well as ko, suicide, taxation and button rules from the options described in Wu (2021). These configuration settings are provided as input to the neural network (Wu, 2019, Table 4), so the network should learn to play appropriately under a range of rule sets. Additionally, during training Komi was sampled randomly from a normal distribution with mean 7 and standard deviation 1 (Wu, 2019, Appendix D).

---

[3]In private communication, the author of KataGo estimated that modified Tromp-Taylor made up a "a few %" of the training data, "growing to more like 10% or as much as 20%" depending on differences such as "self-capture and ko rules that shouldn't matter for what you're investigating, but aren't fully the same rules as Tromp-Taylor".

### A.1   Difference From Typical Human Play

Although KataGo supports a variety of rules, all of them involve automatically scoring the board at the end of the game. By contrast, when a match between humans end, the players typically confer and agree which stones are dead, removing them from the board prior to scoring. If no agreement can be reached then the players either continue playing the game until the situation is clarified, or a referee arbitrates the outcome of the game.

KataGo has a variety of optional features to help it play well under human scoring rules. For example, KataGo includes an auxiliary prediction head for whether stones are dead or alive. This enables it to propose which stones it believes are dead when playing on online Go servers. Additionally, it include hard-coded features that can be enabled to make it play in a more human-like way, such as `friendlyPassOk` to promote passing when heuristics suggest the game is nearly over.

These features have led some to speculate that the (undefended) victim passes prematurely in games such as those in Figure 1 (left) because it has learned or is configured to play in a more human-like way. *Prima facie*, this view seems credible: a human player certainly might pass in a similar situation to our victim, viewing the game as already won under human rules. Although tempting, this explanation is not correct: the optional features described above were disabled in our evaluation. Therefore KataGo loses under the rules it was both trained and configured to use.

In fact, the majority of our evaluation used the `match` command to run KataGo vs. KataGo agents which naturally does not support these human-like game play features. We did use the `gtp` command, implementing the Go Text Protocol (GTP), for a minority of our experiments, such as evaluating KataGo against other AI systems or human players. In those experiments, we configured `gtp` to follow the same Tromp-Taylor rules described above, with any human-like extensions disabled.

## B   Search Algorithms

### B.1   A Review of Monte-Carlo Tree Search (MCTS)

In this section, we review the basic Monte-Carlo Tree Search (MCTS) algorithm as used in AlphaGo-style agents (Silver et al., 2016). This formulation is heavily inspired by the description of MCTS given in Wu (2019).

MCTS is an algorithm for growing a game tree one node at a time. It starts from a tree $T_0$ with a single root node $x_0$. It then goes through $N$ *playouts*, where every playout adds a leaf node to the tree. We will use $T_i$ to denote the game tree after $i$ playouts, and will use $x_i$ to denote the node that was added to $T_{i-1}$ to get $T_i$. After MCTS finishes, we have a tree $T_N$ with $N + 1$ nodes. We then use simple statistics of $T_N$ to derive a sampling distribution for the next move.

#### B.1.1   MCTS Playouts

MCTS playouts are governed by two learned functions:

  a. A value function estimator $\hat{V} : \mathcal{T} \times \mathcal{X} \to \mathbb{R}$, which returns a real number $\hat{V}_T(x)$ given a tree $T$ and a node $x$ in $T$. The value function estimator is meant to estimate how good it is to be at $x$ from the perspective of the player to move at the root of the tree.

  b. A policy estimator $\hat{\pi} : \mathcal{T} \times \mathcal{X} \to \mathcal{P}(\mathcal{X})$, which returns a probability distribution over possible next states $\hat{\pi}_T(x)$ given a tree $T$ and a node $x$ in $T$. The policy estimator is meant to approximate the result of playing the optimal policy from $x$ (from the perspective of the player to move at $x$).

For both KataGo and AlphaGo, the value function estimator and policy estimator are defined by two deep neural network heads with a shared backbone. The reason that $\hat{V}$ and $\hat{\pi}$ also take a tree $T$ as an argument is because the estimators factor in the sequence of moves leading up to a node in the tree.

A playout is performed by taking a walk in the current game tree $T$. The walk goes down the tree until it attempts to walk to a node $x'$ that either doesn't exist in the tree or is a terminal node.[4] At this point the playout ends and $x'$ is added as a new node to the tree (we allow duplicate terminal nodes in the tree).

Walks start at the root of the tree. Let $x$ be where we are currently in the walk. The child $c$ we walk to (which may not exist in the tree) is given by

$$\text{walk}_T^{\text{MCTS}}(x)$$
$$= \begin{cases} \underset{c}{\arg\max} & \bar{V}_T(c) + \alpha \cdot \hat{\pi}_T(x)[c] \cdot \frac{\sqrt{S_T(x)-1}}{1+S_T(c)} & \text{if root player to move at } x, \\ \underset{c}{\arg\min} & \bar{V}_T(c) - \alpha \cdot \hat{\pi}_T(x)[c] \cdot \frac{\sqrt{S_T(x)-1}}{1+S_T(c)} & \text{if opponent player to move at } x, \end{cases} \quad (1)$$

where the argmin and argmax are taken over all children reachable in a single legal move from $x$. There are some new pieces of notation in Eq 1. Here are what they mean:

1. $\bar{V}_T : \mathcal{X} \to \mathbb{R}$ takes a node $x$ and returns the average value of $\hat{V}_T$ across all the nodes in the subtree of $T$ rooted at $x$ (which includes $x$). In the special case that $x$ is a terminal node, $\bar{V}_T(x)$ is the result of the finished game as given by the game-simulator. When $x$ does not exist in $T$, we instead use the more complicated formula[5]

$$\bar{V}_T(x) = \bar{V}_T(\text{par}_T(x)) - \beta \cdot \sqrt{\sum_{x' \in \text{children}_T(\text{par}_T(x))} \hat{\pi}_T(\text{par}_T(x))[x']} \ ,$$

where $\text{par}_T(x)$ is the parent of $x$ in $T$ and $\beta$ is a constant that controls how much we de-prioritize exploration after we have already done some exploration.

2. $\alpha \geq 0$ is a constant to trade off between exploration and exploitation.

3. $S_T : \mathcal{X} \to \mathbb{Z}_{\geq 0}$ takes a node $x$ and returns the size of the subtree of $T$ rooted at $x$. Duplicate terminal nodes are counted multiple times. If $x$ is not in $T$, then $S_T(x) = 0$.

The first term in Eq 1 can be thought of as the exploitation term, and the second term can be thought of as the exploration term and is inspired by UCB algorithms.

### B.1.2 MCTS FINAL MOVE SELECTION

The final move to be selected by MCTS is sampled from a distribution proportional to

$$S_{T_N}(c)^{1/\tau}, \quad (2)$$

where $c$ in this case is a child of the root node. The temperature parameter $\tau$ trades off between exploration and exploitation.[6]

### B.1.3 EFFICIENTLY IMPLEMENTING MCTS

To efficiently implement the playout procedure one should keep running values of $\bar{V}_T$ and $S_T$ for every node in the tree. These values should be updated whenever a new node is added. The standard formulation of MCTS bakes these updates into the algorithm specification. Our formulation hides the procedure for computing $\bar{V}_T$ and $S_T$ to simplify exposition.

### B.2 ADVERSARIAL MCTS: SAMPLE (A-MCTS-S)

In this section, we describe in detail how our Adversarial MCTS: Sample (A-MCTS-S) attack is implemented. We build off of the framework for vanilla MCTS as described in Appendix B.1.

---

[4]A "terminal" node is one where the game is finished, whether by the turn limit being reached, one player resigning, or by two players passing consecutively.

[5]Which is used in KataGo and LeelaZero but not AlphaGo (Wu, 2019).

[6]See `search.h::getChosenMoveLoc` and `searchresults.cpp::getChosenMoveLoc` to see how KataGo does this.

A-MCTS-S, just like MCTS, starts from a tree $T_0$ with a single root node and adds nodes to the tree via a series of $N$ playouts. We derive the next move distribution from the final game tree $T_N$ by sampling from the distribution proportional to

$$S_{T_N}^{\text{A-MCTS}}(c)^{1/\tau}, \quad \text{where } c \text{ is a child of the root node of } T_N. \tag{3}$$

Here, $S_T^{\text{A-MCTS}}$ is a modified version of $S_T$ that measures the size of a subtree while ignoring non-terminal victim-nodes (at victim-nodes it is the victim's turn to move, and at self-nodes it is the adversary's turn to move). Formally, $S_T^{\text{A-MCTS}}(x)$ is the sum of the weights of nodes in the subtree of $T$ rooted at $x$, with weight function

$$w_T^{\text{A-MCTS}}(x) = \begin{cases} 1 & \text{if } x \text{ is self-node,} \\ 1 & \text{if } x \text{ is terminal victim-node,} \\ 0 & \text{if } x \text{ is non-terminal victim-node.} \end{cases} \tag{4}$$

We grow the tree by A-MCTS playouts. At victim-nodes, we sample directly from the victim's policy $\pi^v$:

$$\text{walk}_T^{\text{A-MCTS}}(x) := \text{sample from } \pi_T^v(x). \tag{5}$$

This is a perfect model of the victim *without* search. However, it will tend to underestimate the strength of the victim when the victim plays with search.

At self-nodes, we instead take the move with the best upper confidence bound just like in regular MCTS:

$$\text{walk}_T^{\text{A-MCTS}}(x) := \underset{c}{\text{argmax}} \quad \bar{V}_T^{\text{A-MCTS}}(c) + \alpha \cdot \hat{\pi}_T(x)[c] \cdot \frac{\sqrt{S_T^{\text{A-MCTS}}(x) - 1}}{1 + S_T^{\text{A-MCTS}}(c)}. \tag{6}$$

Note this is similar to Eq 1 from the previous section. The key difference is that we use $S_T^{\text{A-MCTS}}(x)$ (a weighted version of $S_T(x)$) and $\bar{V}_T^{\text{A-MCTS}}(c)$ (a weighted version of $\bar{V}_T(c)$). Formally, $\bar{V}_T^{\text{A-MCTS}}(c)$ is the weighted average of the value function estimator $\hat{V}_T(x)$ across all nodes $x$ in the subtree of $T$ rooted at $c$, weighted by $w_T^{\text{A-MCTS}}(x)$. If $c$ does not exist in $T$ or is a terminal node, we fall back to the behavior of $\bar{V}_T(c)$.

## B.3 Adversarial MCTS: Victim Model (A-MCTS-VM)

In A-MCTS-VM, we propose fine-tuning a copy of the victim network to predict the moves played by the victim in games played against the adversarial policy. This is similar to how the victim network itself was trained, but may be a better predictor as it is trained on-distribution. The adversary follows the same search procedure as in A-MCTS-S but samples from this predictive model instead of the victim. This therefore has the same inference complexity as A-MCTS-S, with slightly greater training complexity due to the need to train an additional network. However, it does require white-box access to the victim.

## B.4 Pass-Alive Defense

Our hard-coded defense modifies KataGo's C++ code to directly remove passing moves from consideration after MCTS, setting their probability to zero. Since the victim must eventually pass in order for the game to end, we allow passing to be assigned nonzero probability when there are no legal moves, *or* when the only legal moves are inside the victim's own pass-alive territory. We also do not allow the victim to play within its own pass-alive territory—otherwise, after removing highly confident pass moves from consideration, KataGo may play unconfident moves within its pass-alive territory, losing liberties and eventually losing the territory altogether. We use a pre-existing function inside the KataGo codebase, `Board::calculateArea`, to determine which moves are in pass-alive territory.

The term "pass-alive territory" is defined in the KataGo rules as follows (Wu, 2021):

> A {maximal-non-black, maximal-non-white} region R is *pass-alive-territory* for {Black, White} if all {black, white} regions bordering it are pass-alive-groups, and all or all but one point in R is adjacent to a {black, white} pass-alive-group, respectively.

| Hyperparameter | Value | Different from KataGo? |
|---|---|---|
| Batch Size | 256 | Same |
| Learning Rate Scale of Hardcoded Schedule | 1.0 | Same |
| Minimum Rows Before Shuffling | 250,000 | Same |
| Data Reuse Factor | 4 | Similar |
| Adversary Visit Count | 600 | Similar |
| Adversary Network Architecture | `b6c96` | Different |
| Gatekeeping | Disabled | Different |
| Auto-komi | Disabled | Different |
| Komi randomization | Disabled | Different |
| Handicap Games | Disabled | Different |
| Game Forking | Disabled | Different |

Table 1: Key hyperparameter settings for our adversarial training runs.

The notion "pass-alive group" is a standard concept in Go (Wu, 2021):

> A black or white region R is a *pass-alive-group* if there does not exist any sequence of consecutive pseudolegal moves of the opposing color that results in emptying R.

KataGo uses an algorithm introduced by Benson (1976) to efficiently compute the pass-alive status of each group. For more implementation details, we encourage the reader to consult the official KataGo rules and the KataGo codebase on GitHub.

## C  HYPERPARAMETER SETTINGS

We enumerate the key hyperparameters used in our training run in Table 1. For brevity, we omit hyperparameters that are the same as KataGo defaults and have only a minor effect on performance.

The key difference from standard KataGo training is that our adversarial policy uses a `b6c96` network architecture, consisting of 6 blocks and 96 channels. This is much smaller than the victim, which uses a `b20c256` or `b40c256` architecture. We additionally disable a variety of game rule randomizations that help make KataGo a useful AI teacher in a variety of settings but are unimportant for our attack. We also disable gatekeeping, designed to stabilize training performance, as our training has proved sufficiently stable without it.

We train at most 4 times on each data row before blocking for fresh data. This is comparable to the original KataGo training run, although the ratio during that run varied as the number of asynchronous selfplay workers fluctuated over time. We use an adversary visit count of 600, which is comparable to KataGo, though the exact visit count has varied between their training runs.

### C.1  CONFIGURATION FOR CURRICULUM AGAINST VICTIM WITHOUT SEARCH

In Section 5.1, we train using a curriculum over checkpoints, moving on to the next checkpoint when the adversary's win-rate exceeds 50%. We ran the curriculum over the following checkpoints, all without search:

1. Checkpoint 127: `b20c256x2-s5303129600-d1228401921` (`cp127`).
2. Checkpoint 200: `b40c256-s5867950848-d1413392747`.
3. Checkpoint 300: `b40c256-s7455877888-d1808582493`.
4. Checkpoint 400: `b40c256-s9738904320-d2372933741`.
5. Checkpoint 469: `b40c256-s11101799168-d2715431527`.
6. Checkpoint 505: `b40c256-s11840935168-d2898845681` (`Latest`).

These checkpoints can all be obtained from Wu (2022).

We start with checkpoint 127 for computational efficiency: it is the strongest KataGo network of its size, 20 blocks or `b20`. The subsequent checkpoints are all 40 block networks, and are approximately equally spaced in terms of training time steps. We include checkpoint 469 in between 400 and 505 for historical reasons: we ran some earlier experiments against checkpoint 469, so it is helpful to include checkpoint 469 in the curriculum to check performance is comparable to prior experiments.

Checkpoint 505 is the latest *confidently rated* network. There are some more recent, larger networks (`b60` = 60 blocks) that may have an improvement of up to 150 Elo. However, they have had too few rating games to be confidently evaluated.

## C.2 Configuration for Curriculum Against Victim with Passing Defense

In Section 5.2, we ran the curriculum over the following checkpoints, all with the pass-alive defense enabled:

1. Checkpoint 39: `b6c96-s45189632-d6589032` ($cp39_{def}$), no search
2. Checkpoint 49: `b6c96-s69427456-d10051148`, no search.
3. Checkpoint 63: `b6c96-s175395328-d26788732`, no search.
4. Checkpoint 79: `b10c128-s197428736-d67404019`, no search.
5. Checkpoint 99: `b15c192-s497233664-d149638345`, no search.
6. Checkpoint 127: `b20c256x2-s5303129600-d1228401921`, no search ($cp127_{def}$).
7. Checkpoint 200: `b40c256-s5867950848-d1413392747`, no search
8. Checkpoint 300: `b40c256-s7455877888-d1808582493`, no search.
9. Checkpoint 400: `b40c256-s9738904320-d2372933741`, no search.
10. Checkpoint 469: `b40c256-s11101799168-d2715431527`, no search.
11. Checkpoint 505: `b40c256-s11840935168-d2898845681` ($Latest_{def}$), no search (1 visit).
12. Checkpoint 505: `b40c256-s11840935168-d2898845681` ($Latest_{def}$), 2 visits.
13. Checkpoint 505: `b40c256-s11840935168-d2898845681` ($Latest_{def}$), 4 visits.
14. Checkpoint 505: `b40c256-s11840935168-d2898845681` ($Latest_{def}$), 8 visits.
15. Checkpoint 505: `b40c256-s11840935168-d2898845681` ($Latest_{def}$), 16 visits.
16–18. ...
19. Checkpoint 505: `b40c256-s11840935168-d2898845681` ($Latest_{def}$), 256 visits.

We move on to the next checkpoint when the adversary's win-rate exceeds 50% until we reach $Latest_{def}$ without search, at which point we increase the win-rate threshold to 75%.

## D Strength of Go AI systems

In this section, we estimate the strength of KataGo's `Latest` network with and without search nd the AlphaZero agent from (Schmid et al., 2021) playing with 800 visits.

### D.1 Strength of KataGo Without Search

First, we estimate the strength of KataGo's `Latest` agent playing without search. We use two independent methodologies and conclude that `Latest` without search is at the level of a weak professional.

One way to gauge the performance of `Latest` without search is to see how it fares against humans on online Go platforms. Per Table 2, on the online Go platform KGS, a slightly earlier (and weaker) checkpoint than `Latest` playing without search is roughly at the level of a top-100 European player. However, some caution is needed in relying on KGS rankings:

| KGS handle | Is KataGo? | KGS rank | EGF rank | EGD Profile |
|---|---|---|---|---|
| Fredda | | 22 | 25 | Fredrik Blomback |
| cheater | | 25 | 6 | Pavol Lisy |
| TeacherD | | 26 | 39 | Dominik Boviz |
| NeuralZ03 | ✓ | 31 | | |
| NeuralZ05 | ✓ | 32 | | |
| NeuralZ06 | ✓ | 35 | | |
| ben0 | | 39 | 16 | Benjamin Drean-Guenaizia |
| sai1732 | | 40 | 78 | Alexandr Muromcev |
| Tichu | | 49 | 64 | Matias Pankoke |
| Lukan | | 53 | 10 | Lukas Podpera |
| HappyLook | | 54 | 49 | Igor Burnaevskij |

Table 2: Rankings of various humans and no-search KataGo bots on KGS (KGS, 2022b). Human players were selected to be those who have European Go Database (EGD) profiles (EGD, 2022), from which we obtained the European Go Federation (EGF) rankings in the table. The KataGo bots are running with a checkpoint slightly weaker than `Latest`, specifically Checkpoint 469 or `b40c256-s11101799168-d2715431527` (Rob, 2022). Per Wu (2022), the checkpoint is roughly 10 Elo weaker than `Latest`.

1. Players on KGS compete under less focused conditions than in a tournament, so they may underperform.

2. KGS is a less serious setting than official tournaments, which makes cheating (e.g., using an AI) more likely. Thus human ratings may be inflated.

3. Humans can play bots multiple times and adjust their strategies, while bots remain static. In a sense, humans are able to run adversarial attacks on the bots, and are even able to do so in a white-box manner since the source-code and network weights of a bot like KataGo are public.

Another way to estimate the strength of `Latest` without search is to compare it to other AIs with known strengths and extrapolate performance across different amounts of search. Our analysis critically assumes the transitivity of Elo at high levels of play. We walk through our estimation procedure below:

1. Our anchor is ELF OpenGo at 80,000 visits per move, which won all 20 games played against four top-30 professional players, including five games against the now world number one (Tian et al., 2019). We assume that ELF OpenGo at 80,000 visits is strongly superhuman, meaning it has a 90%+ win rate over the strongest current human.[7] At the time of writing, the top ranked player on Earth has an Elo of 3845 on goratings.org (Coulom, 2022). Under our assumption, ELF OpenGo at 80,000 visits per move would have an Elo of 4245+ on goratings.org.

2. The strongest network in the original KataGo paper was shown to be slightly stronger than ELF OpenGo (Wu, 2019, Table 1) when both bots were run at 1600 visits per move. From Figure 6, we see that the relative strengths of KataGo networks is maintained across different amounts of search. We thus extrapolate that strongest network in the original KataGo paper with 80,000 visits would also have an Elo of 4245+ on goratings.org.

3. The strongest network in the original KataGo paper is comparable to the `b15c192-s1503689216-d402723070` checkpoint on katagotraining.org (Wu, 2022). We dub this checkpoint `Original`. In a series of benchmark games, we found that `Latest` without search won 29/3200 games against `Original` with 1600 visits. This puts `Original` with 1600 visits ~800 Elo points ahead of `Latest` without search.

4. Finally, log-linearly extrapolating the performance of `Original` from 1600 to 80,000 visits using Figure 6 yields an Elo difference of ~900 between the two visit counts.

[7] This assumption is not entirely justified by statistics, as a 20:0 record only yields a 95% binomial lower confidence bound of a 83.16% win rate against top-30 professional players in 2019. It does help however that the players in question were rated #3, #5, #23, and #30 in the world at the time.

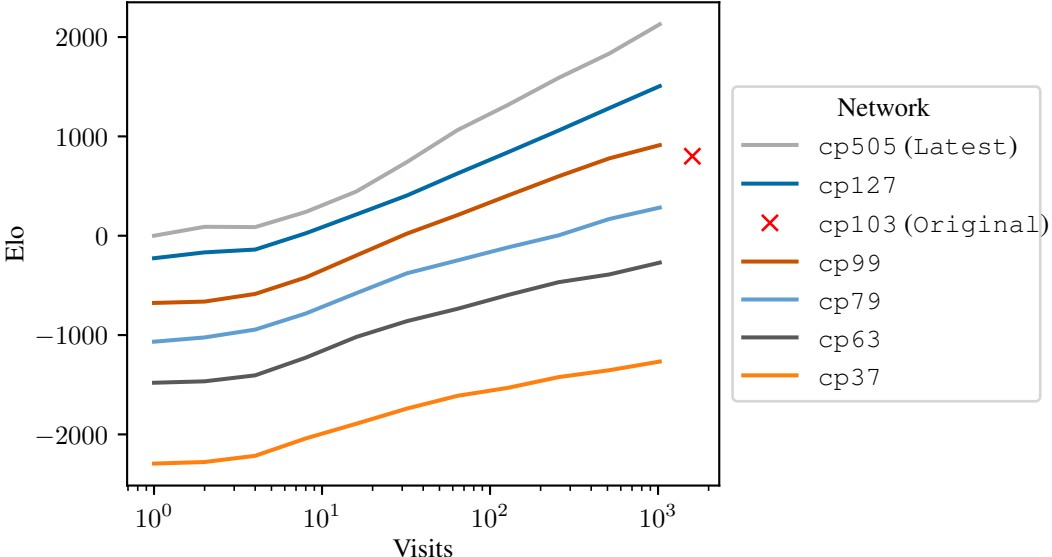

Figure 6: Elo ranking ($y$-axis) of networks (different colored lines) by visit count ($x$-axis). The lines are approximately linear on a log $x$-scale, with the different networks producing similarly shaped lines vertically shifted. This indicates that there is a *consistent* increase in Elo, regardless of network strength, that is logarithmic in visit count. Elo ratings were computed from self-play games among the networks using a Bayesian Elo estimation algorithm (Haoda & Wu, 2022).

5. Combining our work, we get that `Latest` without search is roughly $800 + 900 = {\sim}1700$ Elo points weaker than ELF OpenGo with 80,000 visits. This would give `Latest` without search an Elo rating of $4245 - 1700 = {\sim}2500$ on goratings.org, putting it at the skill level of a weak professional.

As a final sanity check on these calculations, the raw AlphaGo Zero neural network was reported to have an Elo rating of 3,055, comparable to AlphaGo Fan's 3,144 Elo.[8] Since AlphaGo Fan beat Fan Hui, a 2-dan professional player (Silver et al., 2017), this confirms that well-trained neural networks can play at the level of human professionals. Although there has been no direct comparison between KataGo and AlphaGo Zero, we would expect them to be not wildly dissimilar. Indeed, if anything the latest versions of KataGo are likely stronger, benefiting from both a large distributed training run (amounting to over 10,000 V100 GPU days of training) and four years of algorithmic progress.

### D.2 STRENGTH OF KATAGO WITH SEARCH

In the previous section, we established that `Latest` without search is at the level of a weak professional with rating around ~2500 on goratings.org.

Assuming Elo transitivity, we can estimate the strength of `Latest` by utilizing Figure 6. In particular, our evaluation results tell us that `Latest` with 64 playouts/move is roughly 1063 Elo stronger than `Latest` with no search. This puts `Latest` with 64 playouts/move at an Elo of ~3563 on goratings.org—within the top 20 in the world. Beyond 256 playouts/move, `Latest` plays at a superhuman level. `Latest` with 1024 playouts/move, for instance, is roughly 2129 Elo stronger than `Latest` with no search, giving an Elo of ~4629, over 700 points higher than the top player on goratings.org.

---

[8]The Elo scale used in Silver et al. (2017) is not directly comparable to our Elo scale, although they should be broadly similar as both are anchored to human players.

| Agent | Victim? | Elo (rel GnuGo) | Elo (rel victim) |
|---|---|---|---|
| AlphaZero(s=16k, t=800k) | | +3139 | +1040 |
| AG0 3-day(s=16k) | | +3069 | +970 |
| AlphaGo Lee(time=1s) | | +2308 | +209 |
| **AlphaZero(s=800,t=800k)** | ✓ | **+2099** | 0 |
| Pachi(s=100k) | | +869 | -1230 |
| Pachi(s=10k) | | +231 | -1868 |
| GnuGo(l=10) | | +0 | -2099 |

Table 3: Relative Elo ratings for AlphaZero, drawing on information from Schmid et al. (2021, Table 4), Silver et al. (2018) and Silver et al. (2017). s stands for number of steps, time for thinking time, and t for number of training steps.

### D.3 STRENGTH OF ALPHAZERO

Prior work from Timbers et al. (2022) described in Section 2 exploited the AlphaZero replica from Schmid et al. (2021) playing with 800 visits. Unfortunately, this agent has never been evaluated against KataGo or against any human player, making it difficult to directly compare its strength to those of our victims. Moreover, since it is a proprietary model, we cannot perform this evaluation ourselves. Accordingly, in this section we seek to estimate the strength of these AlphaZero agents using three anchors: GnuGo, Pachi and Lee Sedol. Our estimates suggest AlphaZero with 800 visits ranges in strength from the top 200 of human players, to being slightly superhuman.

We reproduce relevant Elo comparisons from prior work in Table 3. In particular, Table 4 of Schmid et al. (2021) compares the victim used in Timbers et al. (2022), AlphaZero(s=800,t=800k), to two open-source AI systems, GnuGo and Pachi. It also compares it to a higher visit count version AlphaZero(s=16k, t=800k), from which we can compare using Silver et al. (2018) to AG0 3-day and from there using Silver et al. (2017) to AlphaGo Lee which played Lee Sedol.

Our first strength evaluation uses the open-source anchor point provided by Pachi(s=10k). The authors of Pachi (Baudiš & Gailly, 2012) report it achieves a 2-dan ranking on KGS (Baudiš & Gailly, 2020) when playing with 5000 playouts and using up to 15,000 when needed. We conservatively assume this corresponds to a 2-dan EGF player (KGS rankings tend to be slightly inflated), giving an EGF Elo of 2200. Assuming transitivity, the victim AlphaZero(s=800,t=800k) would then have an EGF Elo of 4299. The top EGF professional Ilya Shiskin has an EGF Elo of 2830 (Federation, 2022) at the time of writing, and 2979 on goratings.org (Coulom, 2022). Using Ilya as an anchor, this would give AlphaZero(s=800,t=800k) a goratings.org Elo of 4299+2979-2830=4448. This is superhuman, as the top player at the time of writing has an Elo of 3845.

However, some caution is needed here—the Elo gap between Pachi(s=10k) and AlphaZero(s=800,t=800k) is huge, making the exact value unreliable. The gap from Pachi(s=100k) is smaller, however unfortunately to the best of our knowledge there is no public evaluation of Pachi at this strength. However, the results in Baudiš & Gailly (2020) strongly suggest it would perform at no more than a 4-dan KGS level, or at most an EGF Elo of 2400.[9] Repeating the analysis above then gives a goratings.org Elo of 2400+(2308-869)+(2979-2830)=3988 Elo. This still suggests the victim is superhuman, but only barely.

However, if we instead take GnuGo level 10 as our anchor, we get a quite different result. It is known to play between 10 and 11kyu KGS (KGS, 2022a), or an EGF Elo of around 1050. This gives an

---

[9]In particular, Baudiš & Gailly (2020) report that Pachi achieves a 3-dan to 4-dan ranking on KGS when playing on a cluster of 64 machines with 22 threads, compared to 2-dan on a 6-core Intel i7. Figure 4 of Baudiš & Gailly (2012) confirms playouts are proportional to the number of machines and number of threads, and we'd therefore expect the cluster to have 200x as many visits, or around a million visits. If 1 million visits is at best 4-dan, then 100,000 visits should be weaker. However, there is a confounder: the 1 million visits was distributed across 64 machines, and Figure 4 shows that distributed playouts do worse than playouts on a single machine. Nonetheless, we would not expect this difference to make up for a 10x difference in visits. Indeed, Baudiš & Gailly (2012, Figure 4) shows that 1 million playouts spread across 4 machines (red circle) is substantially better than 125,000 visits on a single machine (black circle), achieving an Elo of around 150 compared to -20.

implied EGF Elo for AlphaZero(s=800,t=800k) of 3149, or a goratings.org Elo of 3298 Elo. This is still strong, in the top 200 world players, but is far from superhuman.

The large discrepancy between these results led us to seek a third anchor point: how AlphaZero performed relative to previous AlphaGo models that played against humans. A complication is that the version of AlphaZero that Timbers et al. use differs from that originally reported in Silver et al. (2018), however based on private communication with Timbers et al. we are confident the performance is comparable:

> These agents were trained identically to the original AlphaZero paper, and were trained for the full 800k steps. We actually used the original code, and did a lot of validation work with Julian Schrittweiser & Thomas Hubert (two of the authors of the original AlphaZero paper, and authors of the ABR paper) to verify that the reproduction was exact. We ran internal strength comparisons that match the original training runs.

Table 1 of Silver et al. (2018) shows that AlphaZero is slightly stronger than AG0 3-day (AlphaGo Zero, after 3 days of training), winning 60 out of 100 games giving an Elo difference of +70. This tournament evaluation was conducted with both agents having a thinking time of 1 minute. Table S4 from Silver et al. (2018) reports that 16k visits are performed per second, so the tournament evaluation used a massive $960k$ visits–significantly more than reported on in Table 3. However, from Figure 6 we would expect the *relative* Elo to be comparable between the two systems at different visit counts, so we extrapolate AG0 3-day at 16k visits as being an Elo of $3139 - 70 = 3069$ relative to AlphaZero(s=16k, t=800k).

Figure 3a from Silver et al. (2017) report that AG0 3-day achieves an Elo of around 4500. This compares to an Elo of 3,739 for AlphaGo Lee. To the best of our knowledge, the number of visits achieved per second of AlphaGo Lee has not been reported. However, we know that AG0 3-day and AlphaGo Lee were given the same amount of thinking time, so we can infer that AlphaGo Lee has an Elo of $-761$ relative to AG0 3-day. Consequently, AlphaGo Lee(time=1s) thinking for 1 second has an Elo relative to GnuGo of $3069 - 761 = 2308$.

Finally, we know that AlphaGo Lee beat Lee Sedol in four out of five matches, giving AlphaGo Lee a +240 Elo difference relative to Lee Sedol, or that Lee Sedol has an Elo of 2068 relative to Gnu Go level 10. This would imply that the victim is slightly stronger than Lee Sedol. However, this result should be taken with some caution. First, it relies on transitivity through many different versions of AlphaGo. Second, the match between AlphaGo Lee and Lee Sedol was played under two hours of thinking time with 3 byoyomi periods of 60 seconds per move Silver et al. (2018, page 30). We are extrapolating from this to some hypothetical match between AlphaGo Lee and Lee Sedol with only 1 second of thinking time per player. Although the Elo rating of Go AI systems seems to improve log-linearly with thinking time, it is unlikely this result holds for humans.

## E    DETAILED EVALUATION AGAINST KATAGO WITHOUT PASS-ALIVE DEFENSE

In this section we provide more evaluation of our attack from Section 5.1 against victims without the pass-alive defense.

### E.1    ATTACKING THE VICTIM POLICY NETWORK

In Figure 7 we evaluate our adversarial policy against the policy networks of both `cp127` and `Latest` throughout the training process of our policy. We find our adversary attains a large (>90%) win rate against both victims throughout much of training. However, over time the adversary overfits to `Latest`, with the win rate against `cp127` falling to around 20%.

### E.2    TRANSFERRING TO A VICTIM WITH SEARCH

We evaluate the ability of the adversarial policy trained in the Section 5.1 to exploit `Latest` playing *with* search. The policy was trained only against no-search victims. Although our policy using A-MCTS-S with 200 visits achieves a win rate of 100% over 160 games against `Latest` without

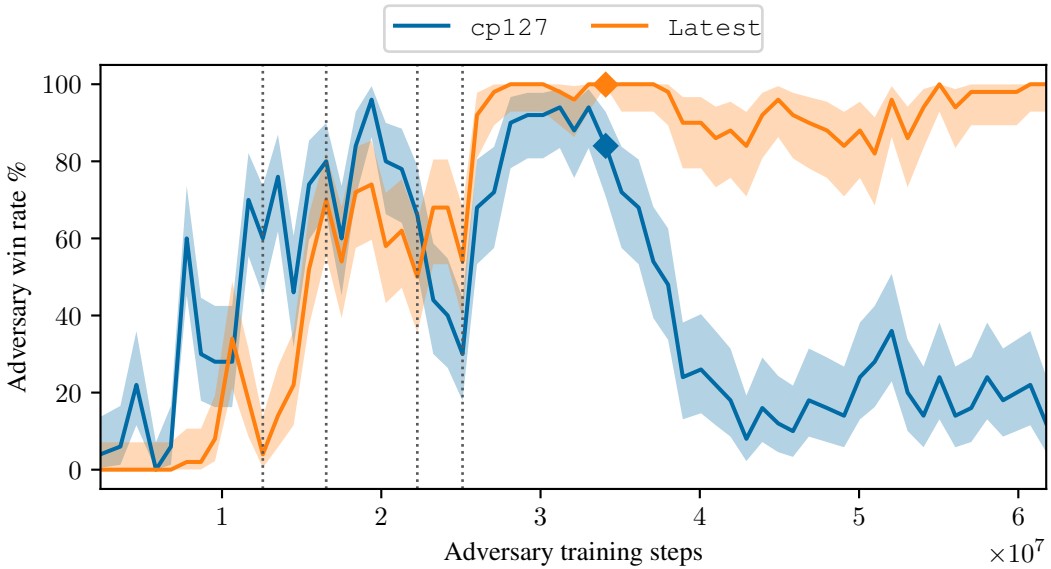

Figure 7: The win rate ($y$-axis) of the adversarial policy over time ($x$-axis) against the `cp127` and `Latest` victim policy networks playing without search. The strongest adversary checkpoint (marked ♦) wins 999/1000 games against `Latest`. The adversary overfits to `Latest`, winning less often against `cp127` over time. Shaded interval is a 95% Clopper-Pearson interval over $n = 50$ games per checkpoint. The adversarial policy is trained with a curriculum, starting from `cp127` and ending at `Latest`. Vertical dashed lines denote switches to a later victim training policy.

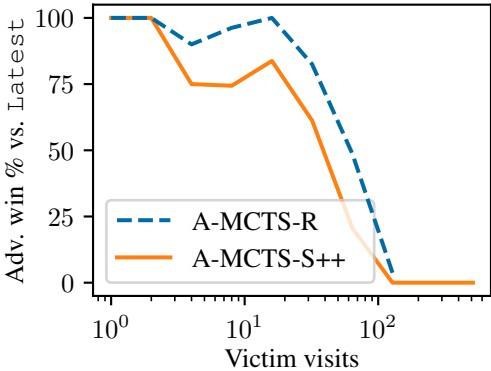

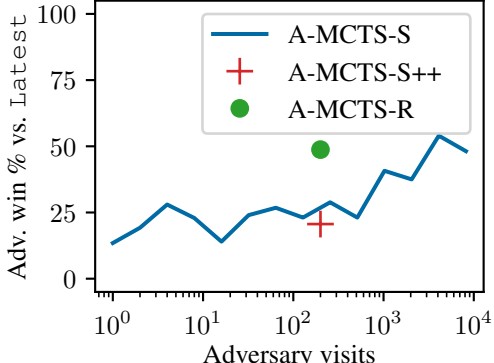

(a) Win rate by number of victim visits ($x$-axis) for A-MCTS-S and A-MCTS-R. The adversary is run with 200 visits. The adversary is unable to exploit `Latest` when it plays with more than 100 visits.

(b) Win rate by number of adversary visits with A-MCTS-S, playing against `Latest` with 64 visits. We see that higher adversary visits lead to moderately higher win rates.

Figure 8: We evaluate the ability of the adversarial policy from Section 5.1 trained against `Latest` without search to transfer to `Latest` with search.

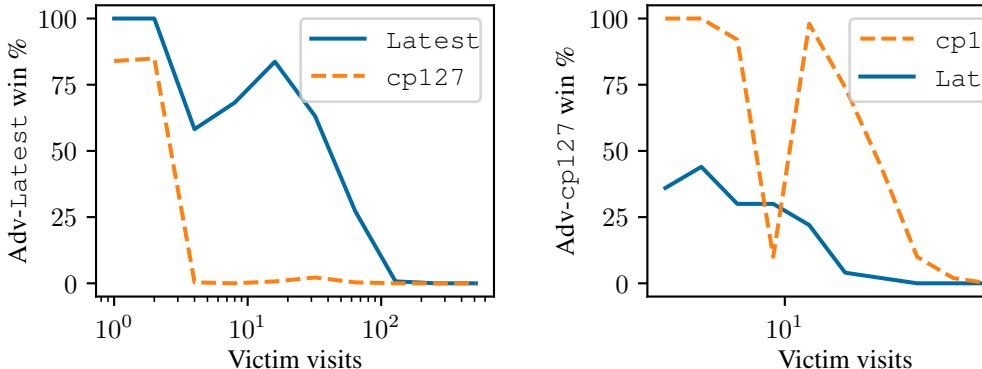

Figure 9: An adversary trained against `Latest` (left) or `cp127` (right), evaluated against both `Latest` and `cp127` at various visit counts. The adversary always uses 600 visits / move.

search, in Figure 8a we find the win rate drops to 61.3% at 32 victim visits. However, A-MCTS-S models the victim as having no search at both training and inference time. We also test A-MCTS-R, which correctly models the victim at inference by performing an MCTS search at each victim node in the adversary's tree. We find that A-MCTS-R performs somewhat better, obtaining a 82.5% win rate against `Latest` with 32 visits, but performance drops to 3.8% at 128 visits.

Of course, A-MCTS-R is more computationally expensive than A-MCTS-S. An alternative way to spend our inference-time compute budget is to perform A-MCTS-S with a greater *adversary* visit count. In Figure 8b we show that we obtain up to a 54% win rate against `Latest` with 64 visits when the adversary has 4,096 visits. This is very similar to the performance of A-MCTS-R with 200 visits, which has a 49% win rate against the same victim. Interestingly, the inference cost of these attacks is also similar, with 4,096 neural network forward passes (NNFPs) per move for A-MCTS-S (one per visit) versus 6,500 NNFPs / move for A-MCTS-R.[10]

Note these experiments only attempt to transfer our adversarial policy. It would also be possible to repeat the attack from scratch against a victim with search, producing a new adversarial policy. We leave this for future work.

### E.3 TRANSFERRING TO OTHER CHECKPOINTS

From Figure 9, we train adversaries against the `Latest` and `cp127` checkpoints respectively and evaluate against both checkpoints. An adversary trained against `Latest` does better against `Latest` than `cp127`, despite `Latest` being a stronger agent. The converse also holds: an agent trained against `cp127` does better against `cp127` than `Latest`. This pattern holds for most visit counts where the adversary wins consistently, although in the case of the adversary for `Latest` the gap is fairly narrow (99% vs. 80% win rate) at low visit counts. These results suggest that different checkpoints have unique vulnerabilities.

### E.4 BASELINE ATTACKS

We also test *hard-coded* baseline adversarial policies. These baselines were inspired by the behavior of our trained adversary. The *Edge* plays random legal moves in the outermost $\ell^\infty$-box available on the board. The *Spiral* attack is similar to the *Edge* attack, except that it plays moves in a deterministic counterclockwise order, forming a spiral pattern. Finally, we also implement *Mirror Go*, a classic novice strategy which plays the opponent's last move reflected about the $y = x$ diagonal. If the opponent plays on $y = x$, Mirror Go plays that move reflected along the $y = -x$ diagonal. If the mirrored vertex is taken, Mirror Go plays the closest legal move by $\ell^1$ distance.

In Figure 10, we plot the win rate of the baseline attacks against the KataGo victim `Latest`, and in in Figure 11, we plot the win margin of `Latest` playing against baselines. The edge attack is the

---

[10]A-MCTS-R with 200 visits performs $100 \cdot 64 + 100 = 6500$ NNFPs / move.

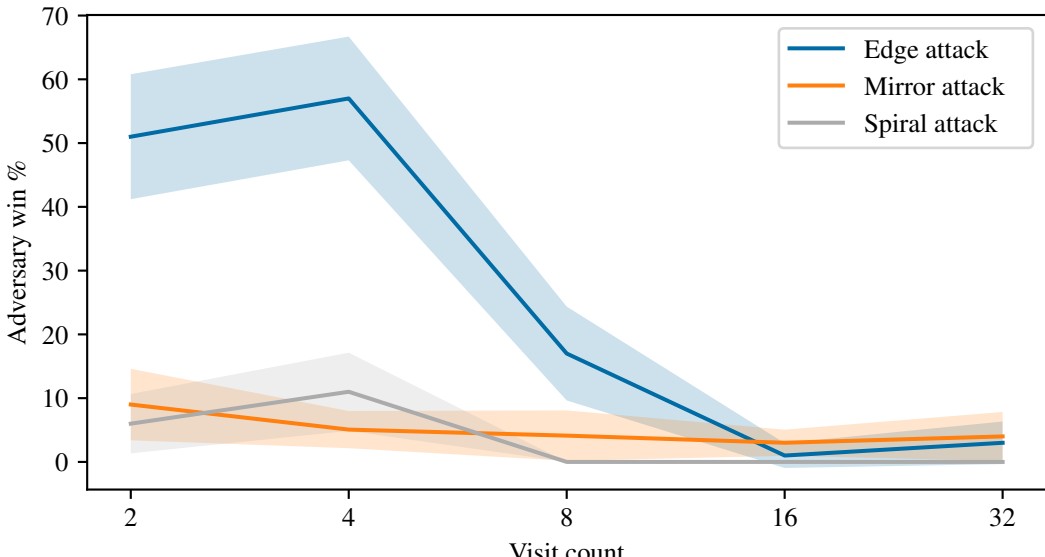

Figure 10: Win rates of different baseline adversaries (see Section 4) versus `Latest` at varying visit counts ($x$-axis) *with adversary playing as white*. 95% CIs are shown. See Figure 11 for the average win margin of KataGo against the baselines.

most successful, achieving over 50% win rate at when `Latest` plays with few visits, but none of the attacks work well once it is playing with at least 16 visits.

### E.5 MIMICKING THE ADVERSARIAL POLICY

Our passing-based adversarial policy appears to follow a very simple strategy. It plays in the corners and edges, staking out a small region of territory while allowing the victim to amass a larger territory. However, the adversary ensures that it is ahead in raw points prior to the victim securing its territory. If the victim then passes prematurely, the adversary wins.

However, it is possible that this seemingly simple policy hides a more nuanced exploit. For example, perhaps the pattern of stones it plays form an adversarial example for the victim's network. To test this, one of the authors attempted to mimic the adversarial policy after observing some of its games.

The author was unable replicate this attack when KataGo was configured in the same manner as for the training and evaluation runs in this paper. However, when the `friendlyPassOk` flag in KataGo was turned on, the author was able successfully replicate this attack against the `NeuralZ06` bot on KGS, as illustrated in Figure 12. This bot uses checkpoint 469 (see Appendix C.1) with no search. The author has limited experience in Go and is certainly weaker than 20 kyu, so they did not win due to any skill in Go.

We observed in Figure 1 that the adversary appears to win by tricking the victim into passing prematurely, at a time favorable to the adversary. In this section, we seek to answer three key questions. First, *why* does the adversary pass even when it leads to a guaranteed loss? Second, is passing *causally* responsible for the victim losing, or would it lose anyway for a different reason? Third, is the adversary performing a *simple* strategy, or does it contain some hidden complexity?

Evaluating the `Latest` victim without search against the adversary from Section 5.1 over $n = 250$ games, we find that `Latest` passes (and loses) in 247 games and does not pass (and wins) in the remaining 3 games. In all cases, `Latest`'s value head estimates a win probability of greater than 99.5% after the final move it makes, although its true win percentage is only 1.2%. `Latest` predicts it will *win* by $\mu = 134.5$ points ($\sigma = 27.9$) after its final move, and passing would be reasonable if it were so far ahead. But in fact it is just one move away from losing by an average of 86.26 points.

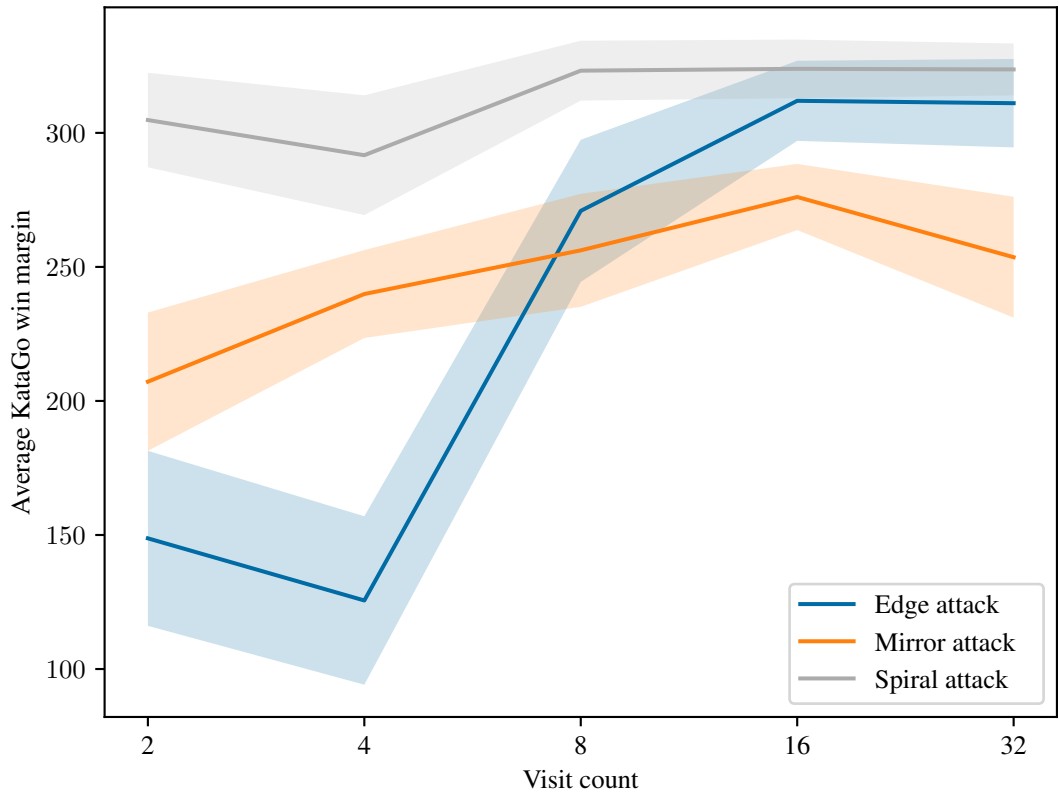

Figure 11: The win margin of the `Latest` *victim* playing against baselines for varying victim visit counts ($x$-axis). Note the margin is positive indicating the victim on average gains more points than the baseline.

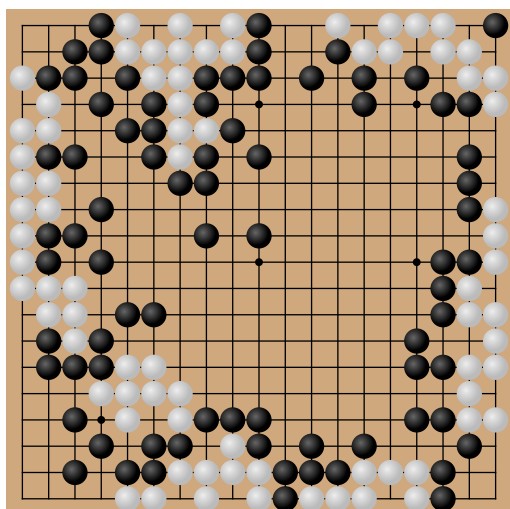

Figure 12: An author of this paper plays as white mimicking our adversarial policy in a game against a KataGo-powered, 8-dan KGS rank bot `NeuralZ06` which has `friendlyPassOk` enabled. White wins by 18.5 points under Tromp-Taylor rules. See the full game.

We conjecture that the reason why the victim's prediction is so mistaken is that the games induced by playing against the adversarial policy are very different to those seen during the victim's self-play training. Certainly, there is no fundamental inability of neural networks to predict the outcome correctly. The adversary's value head achieves a mean-squared error of only 3.18 (compared to 49,742 for the victim) on the adversary's penultimate move. The adversary predicts it will win 98.6% of the time—extremely close to the true 98.8% win rate in this sample.

To verify whether this pathological passing behavior is the reason the adversarial policy wins, we design a hard-coded defense for the victim, the pass-alive defense described in Section 5.2. Whereas the adversarial policy in Section 5.1 won greater than 99% of games against vanilla `Latest`, we find that it *loses* all 1600 evaluation games against `Latest`$_{\text{def}}$. This confirms the adversarial policy wins via passing.

Unfortunately, this "defense" is of limited effectiveness: as we saw in Section 5.1, repeating the attack method finds a new adversary that can beat it. Moreover, the defense causes KataGo to continue to play even when a game is clearly won or lost, which is frustrating for human opponents. The defense also relies on hard-coded knowledge about Go, using a search algorithm to compute the pass-alive territories.

Finally, we seek to determine if the adversarial policy is winning by pursuing a simple high-level strategy, or via a more subtle exploit such as forming an adversarial example by the pattern of stones it plays. We start by evaluating the hard-coded baseline adversarial policies described in Appendix E.4. In Figure 10, we see that all of our baseline attacks perform substantially worse than our trained adversarial policy (Figure 8a). Moreover, all our baseline attacks only win by komi and therefore never win as black. By contrast, our adversarial policy in Section 5.1 wins playing as either color, and often by a large margin (in excess of 50 points).

We also attempted to manually mimic the adversary's gameplay with limited success in Appendix E.5. Although the basics of our adversarial policy seem easy to mimic, matching its performance is challenging, suggesting it may be performing a more subtle exploit.

# F   OTHER EXPERIMENTAL RESULTS

## F.1   HUMANS VS. ADVERSARIAL POLICIES

The same author from Appendix E.5 (strength weaker than 20kyu) played manual games against both the strongest adversary from Figure 3 and the strongest adversary from Figure 7.

In the games against the adversary from Figure 7, the author was able to achieve an overwhelming victory. In the game against the adversary from Figure 3, the author won but with a much smaller margin. See Figure 13 for details.

Our evaluation is imperfect in one significant way: the adversary was not playing with an accurate model of the author (rather it modeled the author as `Latest` with 1 visit). However, given the poor transferability of our adversary to different KataGo checkpoints (see Figure 3 and Appendix E.3), we predict that the adversary would not win even if it had access to an accurate model of the author.

## F.2   ADVERSARIAL BOARD STATE

This paper focuses on training an *agent* that can exploit Go-playing AI systems. A related problem is to find an adversarial *board state* which could be easily won by a human, but which Go-playing AI systems will lose from. In many ways this is a simpler problem, as an adversarial board state need not be a state that the victim agent would allow us to reach in normal play. Nonetheless, adversarial board states can be a useful tool to probe the blindspots that Go AI systems may have.

In Figure 14 we present a manually constructed adversarial board state. Although quite unlike what would occur in a real game, it represents an interesting if trivial (for a human) problem. The black player can always win by executing a simple strategy. If white plays in between two of black's disconnected groups, then black should immediately respond by connecting those groups together. Otherwise, the black player can connect any two of its other disconnected groups together. Whatever the white player does, this strategy ensures that blacks' groups will eventually all be connected

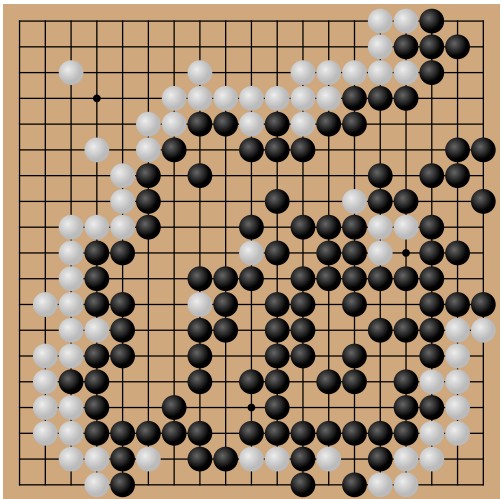

(a) An author (B) defeats the strongest adversary from Figure 3 by 68.5 points. Explore the game.

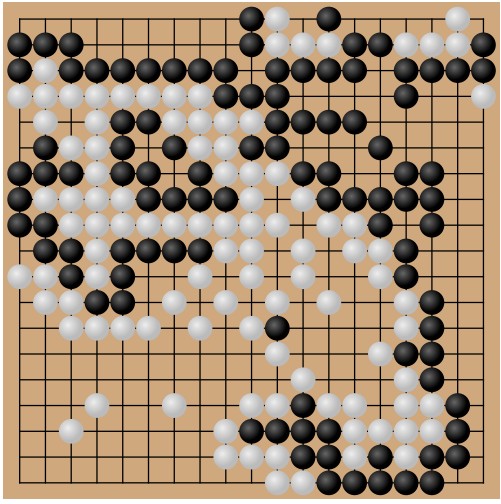

(b) An author (W) defeats the strongest adversary from Figure 3 by 43.5 points. Explore the game.

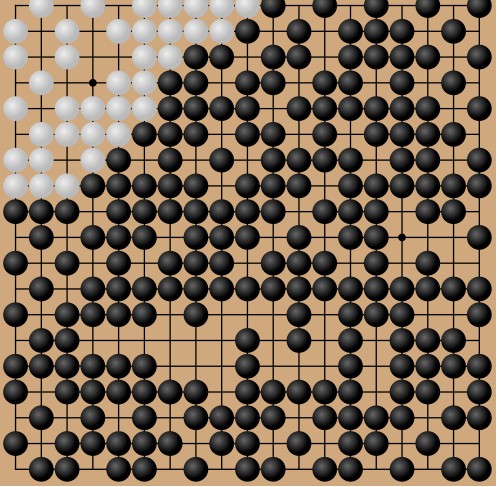

(c) An author (B) defeats the strongest adversary from Figure 7. Explore the game.

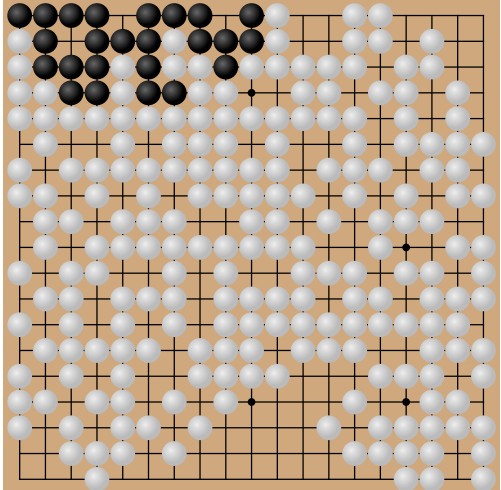

(d) An author (W) defeats the strongest adversary from Figure 7 using A-MCTS-S++. Explore the game.

Figure 13: Games between an author of this paper and the strongest adversaries from Figure 3 and Figure 7. In all games, the author achieves a victory. The adversary used 600 playouts / move and used `Latest` as the model of its human opponent. The adversary used A-MCTS-S for all games except the one marked otherwise.

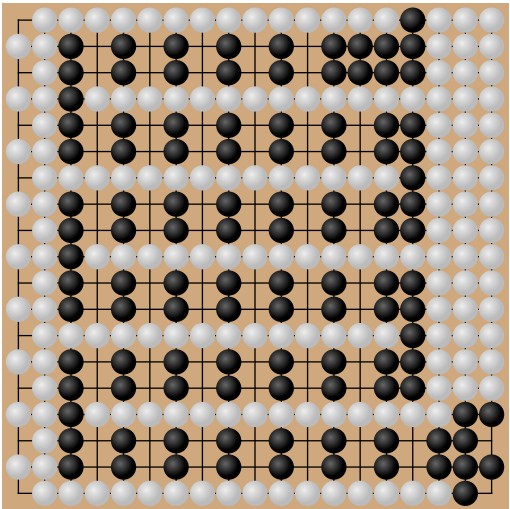

Figure 14: A hand-crafted adversarial example for KataGo and other Go-playing AI systems. It is black's turn to move. Black can guarantee a win by connecting its currently disconnected columns together and then capturing the large white group on the right. However, KataGo playing against itself from this position loses 48% of the time—and 18% of the time it wins by a narrow margin of only 0.5 points!

together. At this point, black has surrounded the large white group on the right and can capture it, gaining substantial territory and winning.

Although this problem is simple for human players to solve, it proves quite challenging for otherwise sophisticated Go AI systems such as KataGo. In fact, KataGo playing against a copy of itself *loses* as black 48% of the time. Even its wins are far less decisive than they should be—18% of the time it wins by only 0.5 points! We conjecture this is because black's winning strategy, although simple, must be executed flawlessly and over a long horizon. Black will lose if at any point it fails to respond to white's challenge, allowing white to fill in both empty spaces between black's groups. This problem is analogous to the classical cliff walking reinforcement learning task (Sutton & Barto, 2018, Example 6.6).

### F.3 HUMAN ANALYSIS OF CAPTURE-TYPE ADVERSARY

In the following we present human analysis of a game played by an adversary of the type shown in Figure 1b. This analysis was done by an expert-level Go player on our team. The game shows typical behavior and outcomes with this adversary: the victim gains an early and soon seemingly insurmountable lead. The adversary sets a trap that would be easy for a human to see and avoid. But the victim is oblivious and collapses.

In this game the victim plays black and the adversary white. The full game is available on our website. We see in Figure 15a that the adversary plays non-standard, subpar moves right from the beginning. The victim's estimate of its win rate is over 90% before move 10, and a human in a high-level match would likewise hold a large advantage from this position.

On move 20 (Figure 15b), the adversary initiates a tactic we see consistently, to produce a "dead" (at least, according to normal judgment) square 4 group in one quadrant of the board. Elsewhere, the adversary plays low, mostly second and third line moves. This is also common in its games, and leads to the victim turning the rest of the center into its sphere of influence. We suspect this helps the adversary later play moves in that area without the victim responding directly, because the victim is already strong in that area and feels confident ignoring a number of moves.

On move 74 (Figure 15c), the adversary begins mobilizing its "dead" stones to set up an encirclement. Over the next 100+ moves, it gradually surrounds the victim in the top left. A key pattern here is that it leads the victim into forming an isolated group that loops around and connects to itself

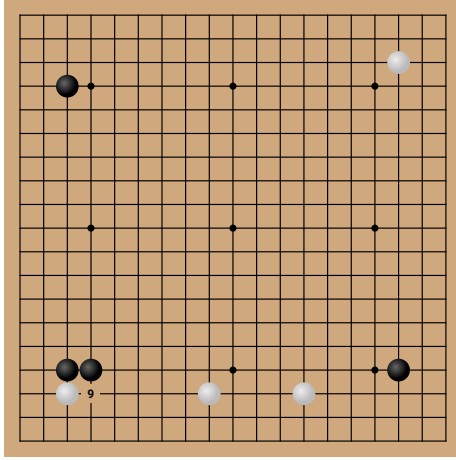

(a) Move 9: after this move victim already has the advantage, if it were robust.

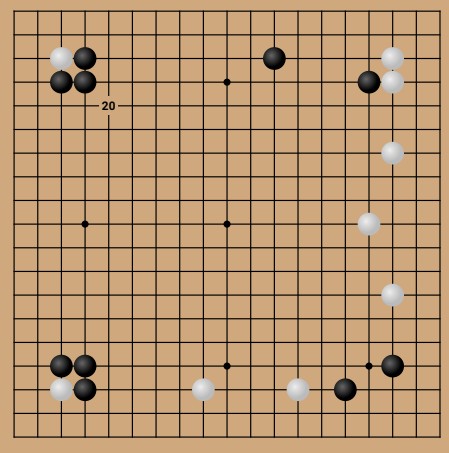

(b) Move 20: adversary initiates a key tactic to create a cycle group.

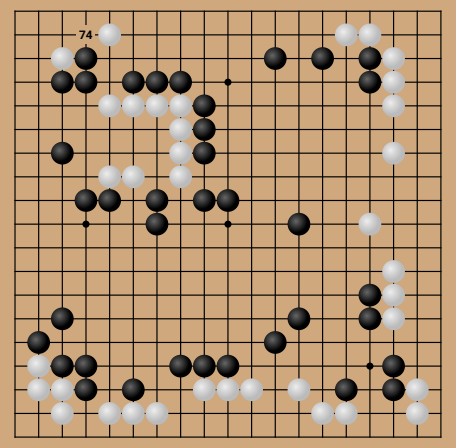

(c) Move 74: adversary slowly begins to surround victim.

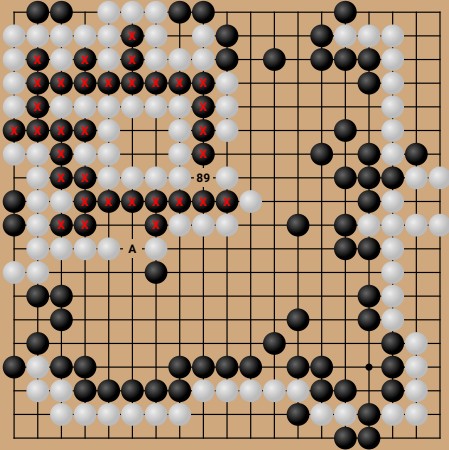

(d) Move 189 (89): victim could have saved X group by playing at "A" instead, but now it will be captured.

Figure 15: An adversarial policy (white) exploiting a KataGo victim (black) by capturing a large group that a human could easily save. The subfigures show different moves in the game. Explore the full game.

(a group with a cycle instead of tree structure). David Wu, creator of KataGo (Wu, 2019), suggested Go-playing agents like the victim struggle to accurately judge the status of such groups, but they are normally very rare. This adversary seems to produce them consistently.

Until the adversary plays move 189 (Figure 15d), the victim could still save that cycle group (marked with X), and in turn still win by a huge margin. There are straightforward moves to do so that would be trivial to find for any human playing at the victim's normal level. Even a human who has only played for a few months or less might find them. For instance, on 189 it could have instead played at the place marked "A." But after 189, it is impossible to escape, and the game is reversed. The victim seems to have been unable to detect the danger. Play continues for another 109 moves but there is no chance for the victim (nor would there be for a human player) to get out of the massive deficit it was tricked into.

