# OpenReview forum: "Adversarial Policies Beat Professional-Level Go AIs"
_ICLR.cc/2023/Conference — Submitted to ICLR 2023_

### Official Review · Reviewer_6zA8 · 2022-10-24

**Confidence:** 5
**Correctness:** 1
**Technical Novelty And Significance:** 2
**Empirical Novelty And Significance:** 1
**Recommendation:** 3

**Clarity, Quality, Novelty And Reproducibility:**

The paper is easy to understand. Due to the complexity of the game programs, the exact results are probably difficult to reproduce exactly, but sufficient details are provided to replicate similar results.

The proposed method is sufficiently novel, but the quality of the results are questionable.


**Strength And Weaknesses:**

Finding winning strategies against high quality playing agents is an interesting topic for at least two reasons: (1) facilitates an improvement of the program, and (2) provides an insight on how strong the state-of-the-art is.

The proposed methods are reasonable, although using gray access presents some limitation on the method (one cannot use it against most commercially available programs).

The weakest point of the paper is the winning strategy obtained. The adversary `tricks' KataGo into stopping the game when there are scattered stones in the territory of KataGo. Such an end position is deemed as a loss for KataGo according to a specific rule used in the experiments, but it would be a clear win according to most ruleset used in human tournaments. In such a tournament if somebody would want to dispute that the stones are alive, it would easy to prove that they are dead. The specific ruleset was designed for computer tournaments to avoid dispute situations, but it is a fairly artificial solution. In fact, it would be annoying to human players to play out games until all dead stones are removed, thus the discovered weakness is in fact a normal functionality of most Go programs.

On a side note, I believe that state-of-the-art Go programs are difficult to exploit not primarily because of their strength, but also because of the stochastic flavor of the MCTS search. A deterministic opponent such as a player using the mean value of the policy network is still a reasonably strong player, but it is easy to combat: just use a full program, find a win, the winning line as adversary (which is a poor standalone player, as such). Alphabeta variants are somewhat less deterministic, but it might still be possible to combat with a similar strategy if the software and hardware are fixed. Against MCTS (unless the seed is fixed), one needs to build a reasonably strong adversary on top of some potentially discovered weaknesses. Alternatively, there can be some `bugs' (rather than weaknesses) in the program that can be exploited, but in my opinion, the issue identified in the paper is a feature that is natural for most Go programs.
The paper presents a method to take advantage of the weaknesses of the Go program KataGo. While the method is fairly elaborate using MCTS and having gray access to KataGo, the winning strategy is not based on superior play, but on the exploitation of a specific rule regarding the termination/scoring of the game.


**Summary Of The Paper:**

The paper presents a method to take advantage of the weaknesses of the Go program KataGo. While the method is fairly elaborate using MCTS and having gray access to KataGo, the winning strategy is not based on superior play, but on the exploitation of a specific rule regarding the termination/scoring of the game.


**Summary Of The Review:**

While the paper is well written, and the proposed method is reasonable, the result (the winning strategy against KataGo) is questionable.

---

> ### Author Response · Authors · 2022-11-11
> **Scoring rules and randomization**
>
> Thank you for your review; we're glad that you find the topic of the paper interesting and the methods reasonable.
>
> **Human vs computer Go scoring rules**
>
> We understand your primary concern is that our adversarial strategy may be exploiting a "normal functionality of most Go programs". Thank you for highlighting this possibility: it was an oversight on our part to not discuss this possibility directly in the paper. We have added appendix A to clarify our evaluation setup, and why we believe this is not the cause of our attack's success. We have also amended the Figure 1 caption to more precisely reflect the evaluation setting and KataGo's training regime.
>
> To summarise appendix A, KataGo was trained under (modified) Tromp-Taylor rules [1]. Although it can be configured to play under Chinese/Japanese rules to avoid being "annoying to human players", in our experiments the KataGo victim is always configured to play under (modified) Tromp-Taylor rules. We chose this to match the evaluation in the KataGo paper (Wu, 2019), and because we would expect the agent to be strongest on the rule set it was trained on.
>
> We acknowledge this setting may seem unnatural from the perspective of human matches. Indeed, we do not intend to claim our results produce an attack for Go under all human rulesets -- just the ruleset that KataGo was trained on and is using. Instead, we view our key contribution as showing that self-play policies trained to achieve some objective X can catastrophically fail at X against adversarially selected opponents, even if they normally perform at a very high level. It is of limited comfort that these policies still do well at some different objective Y, such as a different rule set.
>
> We regret any confusion caused by our inadequate explanation of how KataGo was trained and the rules our KataGo victim was configured to play under. Please do let us know if you find this explanation and our revision to the paper lacking in any way, we would be happy to elaborate on any of these areas.
>
> In response to reviewer t1v2's encouragement, we are running an experiment now to attack a victim hardened with the pass-alive defence described in section 5.4 of the paper. Initial results suggest that this adversary finds an exploit that generalizes across rulesets, as the victim will play the game out to the end, nullifying most major differences in scoring rules. We plan to post a further revision in a few days once we have the final results of this experiment.
>
> **Randomization and MCTS**
> Thanks for making this important point: in a deterministic game like Go, a deterministic victim could be exploited by brute-forcing a sequence of actions that win the game and then replaying those actions. So, paradoxically, randomness makes agents harder to exploit even though optimal play would be deterministic.
>
> However, we're confident our adversary does not win because of this. For one, our adversary achieves a win rate of 54% against victims with 64 visits. This is a small amount of search relative to that typically used in tournament games, but it's enough to introduce significant randomness. Additionally, the state space of Go is huge. So even a small amount of randomness in action selection will quickly lead to a completely unseen board state. Indeed, you can see from the games on https://go-attack-iclr.netlify.app/ that the final board states as well as opening moves vary considerably from game to game.
>
> Nonetheless, the appropriate amount of randomness to use is an interesting question. We used the default KataGo configuration that has already been tuned, but perhaps more randomness (such as a higher softmax temperature to select moves) would increase robustness? We would be happy to run such an experiment if you feel this would strengthen the paper.
>
> [1] The exact rules used for our evaluation of KataGo, and the majority of KataGo's training, are described at https://lightvector.github.io/KataGo/rules.html Click Tromp-Taylor rules and "Enabled" under SelfplayOpts

---

> > ### Comment · Reviewer_6zA8 · 2022-12-08
> > **Rules and bugs**
> >
> > I admit that I have not studied too deeply the training process of KataGo. It is a bit surprising that KataGo fails on the pass issue despite of being trained with the modified rules, but maybe this is an issue of the design of the program, and not of training. Training focuses more on the move to select, or evaluating the position, rather than setting the stoppping-passing rule (which may still be influenced by training, but it depends on external design rules as well).
> >
> > The additional game provided that is generated using the additional pass rule is interesting. It seems to exploit some issue with multipple KO's and adjacent groups that are not safe alone. KO is also a tricky issue in the design of Go problems that are a bit difficult to include in the state representation. The treatment of multiple KO sequences even differ in various rule sets, but I do not think that this is an issue here.
> > I would be interested to see more games, and I think the paper would be improved by providing explanations on the specific/systematic that occur in these games.
> >
> > My understanding of the advesarial attacks presented so far is that it seems to find specific ways to trick KataGo by exploiting certain design choices, rather than finding strategies that would allow to outplay KataGo. However, even if the attacks are limited to exploitation of design choices, the adversarial setting could be helpful to Go programmers to identify and correct design choices that make their program vulnerable.

---

> > > ### Author Response · Authors · 2022-12-08
> > > **Additional example games and some clarifying responses**
> > >
> > > Thank you for taking the time to read our revised version and reply to our comments.
> > >
> > > > I would be interested to see more games, and I think the paper would be improved by providing explanations on the specific/systematic that occur in these games.
> > >
> > > We provide ten examples of games with the new adverssary on the accompanying [anonymized website](https://go-attack-iclr.netlify.app/#2048_visits_hardened) against both KataGo Latest with 2048 visits (at which point it is superhuman) and without search (at which point it is top-100 European). We would be happy to provide more example games if that is of interest to you.
> > >
> > > We provide a detailed qualitative analysis of one game [here](https://go-attack-iclr.netlify.app/game-analysis#qualitative) and in Appendix F.3 of the manuscript. We will aim to more clearly highlight this analysis (currently only linked to from caption of Figure 1) in the camera-ready. Do let us know if there is any additional analysis you'd like to see.
> > >
> > > > It seems to exploit some issue with multipple KO's and adjacent groups that are not safe alone. KO is also a tricky issue in the design of Go problems that are a bit difficult to include in the state representation. The treatment of multiple KO sequences even differ in various rule sets, but I do not think that this is an issue here.
> > >
> > > You're quite right the games often involve multiple KO's. We suspect the adversary may have learned to create these KO's in order to make search less effective, as the victim must search deeper to understand the board state and therefore realize the peril it is in. Section A.1 (page 17) of the [KataGo paper](https://arxiv.org/pdf/1902.10565.pdf) describes the inputs to the network, which include:
> > >   - A channel for whether moving is illegal due to ko/superko.
> > >   - The last 5 move locations, one-hot.
> > >   - Ko rules in use (simple, positional, situational).
> > > We think this state representation includes all the relevant information needed by the victim to evaluate the KO situation accurately, although of course more complex situations will be more difficult for the network to judge.
> > >
> > > > Training focuses more on the move to select, or evaluating the position, rather than setting the stoppping-passing rule (which may still be influenced by training, but it depends on external design rules as well).
> > >
> > > To clarify, KataGo treats passing as one of the possible moves, and it is evaluated by the network and MCTS in the same way as for other possible moves. There is some additional optional hard-coded logic to encourage "friendly" passing when playing against humans, but this is disabled both during training and for our evaluation. We suspect KataGo is vulnerable to this passing attack because it is not exploited during self-play, both because (a) playing against itself is unlikely to create board states that trigger this; (b) if such a board state did arrive, both agents would (wrongly) believe that the victim would win if victim and opponent both pass, so the self-play opponent would tend to choose not to pass.
> > >
> > > > My understanding of the advesarial attacks presented so far is that it seems to find specific ways to trick KataGo by exploiting certain design choices, rather than finding strategies that would allow to outplay KataGo. However, even if the attacks are limited to exploitation of design choices, the adversarial setting could be helpful to Go programmers to identify and correct design choices that make their program vulnerable.
> > >
> > > We agree our method can be used to adversarially test Go programs enabling improvements in the design. However, we believe these issues are ubiquitous with current systems, and are not attributable to any specific design choices. In particular, we found two adversaries winning in qualitatively different ways. Moreover, concurrent work by [Timbers et al](https://arxiv.org/abs/2004.09677) were able to exploit an AlphaZero agent that was trained using an entirely different codebase and input features. Although we certainly hope there are changes to the design of systems that would resolve these problems, it is currently far from clear what they are, and we suspect they are likely to require major modifications to systems (e.g. use PSRO instead of self-play).
> > >
> > > We should clarify that we are not trying to "outplay KataGo" in the sense of building a generally stronger Go program. Indeed, building stronger Go programs may not in itself result in any meaningful scientific insight -- one can quite easily increase the size of KataGo's network and continue the self-play training for longer to produce an agent that will outplay the current version of KataGo. We instead are seeking to test whether narrowly superhuman systems are robust to adversarial attack (we find they are not), to better understand under what circumstances they fail (we find specific, human-understandable patterns like cyclic groups of stones), and how to protect against it (we find search helps but does not fully address the problem).

---

> > > > ### Comment · Reviewer_6zA8 · 2022-12-08
> > > > **clarifying**
> > > >
> > > > KO: I know that the KO point is included in the state representation, but having multiple KOs make things more difficult, because of repetition and so on.
> > > >
> > > > outplay: I meant winning by exhibiting superiority, which in this framework would be limited to certain type of positions/shapes. As an example, in pre-DEEP BLUE chess programs, a useful strategy was to close the position, where there was little need for tactics, but rather long term strategic planning (which was a weakness of programs at that time). A similar case was in Go around the breakthrough results, when the top programs were already strong, but had some difficulty in assessing the viability of large groups.
> > > >
> > > > One could of course categorize weakness with passing and KO-play as strategic weakness, but these are very special features of the Go game: one introduced to shorten the game, when the result is 'clear', and the other to avoid repetitions. Forcing a bit the chess parallel, one could add a draw offer or resignation as a move in chess, and add the field that results in a 3-fold repetition as feature, but these would be a bit awkward, and it is clear that these require special handling. That is not to say that adding the KO intersection is not useful in Go state representation, but it just does not solve the issue completely.

---

> > > > > ### Author Response · Authors · 2022-12-08
> > > > > **Discussion on ko**
> > > > >
> > > > > > KO: I know that the KO point is included in the state representation, but having multiple KOs make things more difficult, because of repetition and so on.
> > > > >
> > > > > It indeed makes search more difficult - both programs and humans have to keep track of more things. However, we do not think KO alone is sufficient make KataGo (or other strong programs) fail. It is intuitive and also a common idea in the Go community that creating more complications might create higher chance of a program making a mistake, but in the past this alone has not produced any strategies that cause significant (game-losing) mistakes consistently.
> > > > >
> > > > > > outplay: I meant winning by exhibiting superiority, which in this framework would be limited to certain type of positions/shapes. As an example, in pre-DEEP BLUE chess programs, a useful strategy was to close the position, where there was little need for tactics, but rather long term strategic planning (which was a weakness of programs at that time). A similar case was in Go around the breakthrough results, when the top programs were already strong, but had some difficulty in assessing the viability of large groups.
> > > > >
> > > > > Although it may be challenging to give a conclusive, quantifiable definition, our adversary does seem to exhibit superiority in regards to cyclic groups. Depending on version, even with substantial search the adversary can achieve a >>50% winrate, which in our analysis so far is based on not only producing but then exploiting ("outplaying") around these cyclic groups. We saw going from chess to Go that a new architecture was needed, due to factors such as the long term strategic planning requirements that you mention (which can be even more extreme in Go, since games often have many more moves). Among other benefits, we hope that finding new weaknesses, such as this cyclic one, may similarly lead to new architectures that outplay and in general are more robust compared to existing ones.
> > > > >
> > > > > > One could of course categorize weakness with passing and KO-play as strategic weakness, but these are very special features of the Go game: one introduced to shorten the game, when the result is 'clear', and the other to avoid repetitions. Forcing a bit the chess parallel, one could add a draw offer or resignation as a move in chess, and add the field that results in a 3-fold repetition as feature, but these would be a bit awkward, and it is clear that these require special handling. That is not to say that adding the KO intersection is not useful in Go state representation, but it just does not solve the issue completely.
> > > > >
> > > > > We agree that ko rules introduce additional complexity into the game of Go, but we would argue it is much more central to Go than passing. As you say, you could avoid passing just by letting players play until all territory is pass-alive, which would be tedious but feasible. Whereas a game like Go fundamentally requires a ko rule to prevent infinite games. In addition, it's an integral part of the game as played by humans, important enough to have books dedicated to it such as as [Fighting Ko by Jin Jiang](https://senseis.xmp.net/?FightingKoTheBook) and the Ko Dictionary by [Murashima Yoshinori](https://senseis.xmp.net/?MurashimaYoshinori). By contrast, we are not aware of any books dedicated to when to pass! If the adversary is better at playing ko positions than the original KataGo victim, we would argue it truly is outplaying KataGo. Additionally, we expect fixes to this vulnerability to result in strengthening KataGo and thereby making it more useful as an analysis tool for ko positions.

---

> > > > > ### Author Response · Authors · 2022-12-11
> > > > > **Additional comment on kos**
> > > > >
> > > > > We the authors want to clarify one more point regarding kos. We wrote earlier that
> > > > >
> > > > > > You're quite right the [adversary] games often involve multiple KO's.
> > > > >
> > > > > After looking through more of our adversary games, we want to amend this statement. Many of our adversary games have at most one active ko through the entire game leading up to and including the time the victim makes its big blunder. See for example: the [game shown in Figure 1b of the paper](https://go-attack-iclr.netlify.app/?row=2#2048_visits_hardened-board), the [fully analyzed game on our website](https://go-attack-iclr.netlify.app/game-analysis), this [game](https://go-attack-iclr.netlify.app/?row=3#2048_visits_hardened-board), or this [other game](https://go-attack-iclr.netlify.app/?row=4#2048_visits_hardened-board).
> > > > >
> > > > > Though, there are indeed some games where there are multiple kos around the time of the victim blunder, like [this one](https://go-attack-iclr.netlify.app/?row=1#2048_visits_hardened-board).

---

> ### Author Response · Authors · 2022-11-11
> **Response to misc points**
>
> In addition to your main concerns regarding the rules and randomness that we replied to in our previous comment, you also highlighted several other concerns:
>
> > The winning strategy is not based on superior play, but on the exploitation of a specific rule regarding the termination/scoring of the game.
>
> We agree the adversary does not win based on superior play, and indeed our own evaluation shows that a human amateur can easily beat the adversary. Indeed, we think this is the interesting part of our result -- that the best response found by training against a KataGo victim is not a generally strong agent, but one that exploits a specific weakness in KataGo.
>
> > 1: The main claims of the paper are incorrect or not at all supported by theory or empirical results.
>
> The only correctness issue we see highlighted in this review is our lack of justification for the use of the Tromp-Taylor rule set. Thank you again for bringing this to our attention. Now that we have addressed this by the addition of appendix A, would you consider amending this rating? If there are any outstanding correctness issues, we would appreciate it if you could bring them to our attention. We would like to ensure we communicate our results clearly and correctly to the scientific community.
>
> Thank you in advance for your time and do not hesitate to ask for any further clarification.

---

### Official Review · Reviewer_7dnb · 2022-10-25

**Confidence:** 5
**Clarity, Quality, Novelty And Reproducibility:** The paper has novel and significant c…
**Correctness:** 4
**Technical Novelty And Significance:** 4
**Empirical Novelty And Significance:** 4
**Recommendation:** 6

**Strength And Weaknesses:**

Strength:

I believe the paper uncovered a super important result that illustrates the security concerns in modern RL-based systems using Go as an example. The result is significant and novel.

Weaknesses:

I don't see any major weakness with the paper. However, I would like to point out that the results in this paper, although significant, are not surprising to me at all. People have seen similar cases in a variety of other machine learning subareas. The reason for this kind of vulnerability is also obvious --- the Go AI system is too complex, and although the system can show very good performance when facing humans (where normal states/inputs are expected), it does not guarantee the same performance when abnormal states are encountered (e.g., those generated by adversarial policies).

Overall, I think the paper has made a significant and novel contribution to the intersection area of reinforcement learning and adversarial machine learning.

**Summary Of The Paper:**

This paper studied how to train an adversarial policy that can beat the strongest AI Go-player KataGo system. This is a purely empirical study that discovered unknown security vulnerabilities hidden in AI Go systems. Interestingly, the results in this paper lead to an ironic conclusion --- while the AI Go-playing system is seemingly professional from a human perspective, it really is not a rational player and can make unprofessional moves when facing a strategic adversary Go player, in this case, another AI Go player that employs an adversarial policy. Therefore, the paper has uncovered a very important negative result in designing efficient and robust AI systems.

**Summary Of The Review:**

I work in related areas.

---

> ### Author Response · Authors · 2022-11-11
> **Reply**
>
> Thank you for the review, and we are glad that you found the result to be significant and novel.
>
> We personally were also not surprised that an exploit was possible. The theoretical deficiencies of self-play are well-known, and we have seen neural networks are usually vulnerable to some form of adversarial attack. However, self-play has been practically very successful, leading many to place perhaps more faith in that is warranted. Even if knowledgeable multi-agent RL researchers would expect this, we believe there is value in a clear empirical demonstration to those who were not yet convinced, in order to create common knowledge that this is a problem needing to be solved.
>
> Do let us know if you have any questions about the paper, or suggestions for experiments that would strengthen the contributions of the paper.

---

### Official Review · Reviewer_t1v2 · 2022-10-25

**Confidence:** 4
**Correctness:** 3
**Technical Novelty And Significance:** 2
**Empirical Novelty And Significance:** 2
**Recommendation:** 5

**Clarity, Quality, Novelty And Reproducibility:**

*Clarity and Reproducibility*:
The paper is overall clearly written and relatively easy to follow. Of course, since the paper focuses on Go, a certain level of familiarity with Go is expected from a reader. The paper seems to provide a sufficient description of the experimental procedure and setup in the main part of the paper, with additional details reported in the appendix.

---
*Quality*:
I believe that this work provides a good starting point for studying adversarial policies in games like Go, but it's results could be further developed, and include different domains, not just Go. This would significantly improve the quality of the results, and demonstrate that these result generalize to other domains.

---
*Novelty/Originality*:
In terms of adversarial setting, I found the novelty/originality of this work quite limited. As I mentioned above, similar results have already reported by prior work, but in different environments. Hence, the main contribution appears to be mostly related to replicating these findings in the game of Go.


**Details Of Ethics Concerns:**

As I wrote above, in the paragraph before section 6, the authors make claims based on the outcome of games that one of the authors played against a Go engine. This does not seem to be a proper scientific study, but since it does involve a human subject, it may need to be subject to an ethical review.

**Strength And Weaknesses:**

**Strengths**

The paper showcases that a sophisticated AI-based Go engines can be susceptible to adversarial attacks that exploit weaknesses of their training procedures, i.e., when a minmax solution or  a Nash equilibrium is not reached with self-play training strategies. This result also has implications for AI systems based on search techniques, indicating that they may be vulnerable to similar security threats. To my understanding, prior works on adversarial policies have primarily considered other environments in their experiments (e.g., MuJoCo, Starcraft). Hence, the paper contributes to this literature by extending the experimental findings of these works to the setting of Go.

---
**Weaknesses**

While I find the results that support the aforementioned findings interesting, I also feel that the paper has several weaknesses, listed below.
- In my opinion the main weakness of this work is novelty/origninality - it's not clear to me what is the conceptual novelty of this work. Most of the results have similar if not the same flavor to those from prior work on adversarial policies. While it is generally interesting to see that these results also generalize to Go, I feel like the paper could have focused not just on Go, but also other similar games, e.g., chess or shogi. As the authors suggest in section 6, some of their findings may not generalize to other games.
- As for the results, I also feel that the current set of results could be extended. For example, based on the paragraph before section 5.3, it seems that the experiment do not test the quality of results when an adversarial policy is trained to attack a victim with search.  Moreover, the experiments do not appear to include the case when an adversarial policy is trained to attack  a victim that has the simple defense from section 5.4. It would be great if the authors could comment on these points.  In general, more results that study robustness question would make the results more significant, especially as robustness techniques against adversarial policies may need to be domain dependent. E.g., Gleave et 2020 include results related to adversarial training, so adding analogous results for this setting would be useful.
- There are also claims in the paper which are not fully supported by data, e.g., the claims in the paragraph before section 6 don't seem to be supported by data obtained through a proper human-subject experiment. In particular, the claims in this paragraph are made based on an experiment in which one of the authors of the paper was playing against two bot players; this is not a proper scientific study.

**Summary Of The Paper:**

The paper studies attack against a SOTA Go-playing AI system. In particular, it considers an attack model based on adversarial policies, and KataGo engine. The paper shows that a frozen KataGo victim is highly susceptible to adversarial policies: an adversarial policy trained with only  0.3% of compute time used for training the victim wins against the victim more than 99% of the time when the victim does not use search, and more than 80% of the time when the victim uses enough search to surpass human-level performance. The authors argue that the corresponding adversarial policy applies a counterintuitive strategy, which would fail agains human players.

**Summary Of The Review:**

Overall, the paper studies an interesting topic, and the type of experimental results it reports are generally important for understand security aspects of AI systems that are to be deployed in safety-critical domains.  That said, the current set of results could be extended and further improved, especially since the novelty of this paper appears to be limited.

---

> ### Author Response · Authors · 2022-11-11
> **Novelty & extensions to work**
>
> Thank you for your detailed review: we appreciate the time you took to review our work and suggest improvements. We have started a training run against a victim with the defence described in section 5.4 as you suggested: preliminary results are promising, we will report back in a few days once we have the final results. We also plan to train against a victim with search: we had avoided doing so as the naive approach drastically increases compute requirements, but we have developed an improved method (A-MCTS-VM) described in section 4 of our revised paper that overcomes this issue.
>
> **Novelty**
>
> We understand your main concern with this work is the novelty: namely, that adversarial policies have already been found in two other environments (MuJoCo and Starcraft). We would agree that merely finding that adversarial policies exist in a third environment would be a marginal contribution. However, we view our key contribution not as exploiting policies in Go, but in exploiting policies that are as strong as top human professionals. By contrast, prior work such as Gleave et al (2020) in MuJoCo and Guo et al (2021) in Starcraft has only attacked agents well below the human baseline. Additionally, the victims we exploit were  trained via AlphaZero-style training that distills MCTS search into a policy, instead of policy-gradient RL algorithms such as PPO. This confirms adversarial policies are a general phenomenon, not an artefact of a specific RL algorithm.
>
> Prior work therefore left open the hope that policies might gain human-level robustness when they reach human-level performance in the average case. Our results show that this is unfortunately not the case. We believe it is important for the community to know that adversarial policies require algorithmic progress to solve, not merely scaling existing techniques.
>
> Concretely, Gleave et al (2020) considered agents that were state-of-the-art for Humanoid continuous control -- but the state-of-the-art there still lags far behind human performance, as the videos of "Normal vs Normal" agents demonstrate. Similarly, the victim StarCraft II agents from Guo seem well below the performance of top professionals, which is unsurprising given that AlphaStar (Vinyals et al, 2019) has yet to be publicly replicated.
>
> **Extensions to the work**
>
> Thank you for your suggestions for extensions. We agree that exploring other games such as chess or shogi is an important direction for future work. We are considering exploring that in a follow-up paper, but we feel this is out of scope for this paper. In particular, there is no reliable open-source replica of AlphaZero in shogi: AobaZero is still undergoing training and has yet to be rigorously evaluated. Chess is more tractable, as Leela Chess Zero is well tested, but would still require a complete rewrite of our attack due to the different codebase.
>
> We also agree with your suggestions for additional Go experiments to run. In light of your advice, we have started training an adversary against a version of our victim with the simple defence (avoiding passing) from section 5.4, and will report back once we have results.
>
> We had hitherto not trained an adversary against a victim with search as our A-MCTS-S attack method models the victim as not performing search. Although we could nonetheless train against a victim with search, we expected the resulting model misspecification to limit performance, and we got reasonably good performance from just transferring an adversary to victims with search (Figure 4(a) and 4(b) in the paper). However, we have recently developed a method we dub A-MCTS-VM and described in the revised version of the paper that resolves this model misspecification by training a network to predict the victim with search. We are currently running an experiment using this technique and will update you once we have results.
>
> **References**
>
> Wenbo Guo, Xian Wu, Sui Huang, Xinyu Xing (2021). Adversarial Policy Learning in Two-player Competitive Games. ICML.
> Adam Gleave, Michael Dennis, Cody Wild, Neel Kant, Sergey Levine, Stuart Russell (2020). Adversarial Policies: Attacking Deep Reinforcement Learning. ICLR.
> Oriol Vinyals, Igor Babuschkin, Wojciech M. Czarnecki, Michaël Mathieu, Andrew Dudzik, and others (2019). Grandmaster level in StarCraft II using multi-agent reinforcement learning. Nature.

---

> ### Author Response · Authors · 2022-11-11
> **Justification of claims**
>
> In addition to the concerns about novelty, and suggestions for future work, that we addressed in our previous comment, we understand you also felt that some of our claims were insufficiently justified.
>
> We agree the games played against the adversary and KataGo victim by one of our authors do not justify any general claims about human Go players. We intended to make only two narrow claims: that (a) the adversary is much weaker than a professional Go player, as even a weak Go amateur can beat it; (b) what the adversary is doing can in part be mimicked by a human. If you feel our current phrasing implies more than this please do let us know and we would be happy to amend it.
>
> We think our claim (a) is well substantiated: the human wins by a huge >250 point margin against KataGo, and the games in https://go-attack-iclr.netlify.app/human-evaluation#amateur_vs_adv do not show any sophisticated game play on the part of the human evaluator. If you have concerns regarding this evaluation we would be happy to run a human-subjects experiment for the camera-ready, but we are unlikely to be able to perform this in the rebuttal period. Alternatively, we could set up a bot on an online Go server enabling you to try playing against it yourself.
>
> Claim (b) we intended as more of a speculative claim. Indeed, our results are somewhat equivocal here: our author wins against the bot in one configuration, but not another. We felt this question was important enough that it was worth including this data point even if it is more suggestive than definitive. However, we would be happy to add more caveats here if you feel it is overly strongly stated, or move it to an appendix to de-emphasize the result until we can investigate it further.
>
> It is important to us that we clearly communicate our results to the scientific community. Are there any other areas where you felt our claims were insufficiently substantiated? We would be happy to revise our paper to address this and other concerns you have.

---

> > ### Comment · Reviewer_t1v2 · 2022-12-06
> > **Thank you for your response**
> >
> > I would like to thank the authors for their detailed response. Additional experiments address some of my concerns, but not all - I will raise my score by one point to reflect this. I remain to believe that the results could be further improved/extended, especially those that that involve a human player - in my opinion, this part requires a proper human subject experiment.

---

### Author Response · Authors · 2022-11-11
**Paper revision v1**

In addition to our responses to individual reviewers, we wanted to summarize the changes made in the paper revision:
  1. We have added appendix A describing the exact rules we used to evaluate our adversary, how these differ from those used in human play but are consistent with the rules used during KataGo training, and how KataGo was configured to match these rules during our evaluation.
  2. We include a link to an (anonymized) website that allows readers to browse a collection of games involving our adversary and the KataGo victim. This provides more games than were originally listed in the paper, enabling the reader to get a sense of how much they vary, and provides an improved user interface for navigating the games.
  3. We updated our related work to add the concurrent work by Lan et al (2022) which was not available when we first made the submission. Lan et al show that KataGo is vulnerable to an adversary adding two new stones to the board, even if the boards do not meaningfully change the outcome of the game from the perspective of a human player. We also updated our discussion of Timbers et al (2022) in response to a revised version of their paper on arXiv that includes a more comprehensive strength evaluation.
  4. We have expanded our previous discussion on defences to include a wider range of algorithms than just adversarial training and population-based training. In particular, we now discuss multi-agent RL algorithms such as counterfactual regret minimization, DeepNash and policy-space response oracles.

At t1v2's suggestion we are currently running an experiment to attack a victim that has been patched to not pass prematurely, thereby protecting against our original adversarial policy. Our preliminary results indicate this adversary achieves a 98.9% win rate against the hardened victim without search. It wins by tricking the victim into forming a large group with few liberties (empty stones adjacent to it, needed for survival) and then eventually killing it. In contrast to the original attack presented in the paper, this adversary would also win under human rules. Here's one [example game](http://eidogo.com/#3OgrjuFnX), with the white victim blundering at move W240 with an auto-atari allowing the black adversary to capture a large group at B241. We will post a further revision of the paper early next week including a full analysis of this new adversary similar to that conducted for our original adversary.

Li-Cheng Lan, Huan Zhang, Ti-Rong Wu, Meng-Yu Tsai, I-Chen Wu, Cho-Jui Hsieh (2022). Are AlphaZero-like Agents Robust to Adversarial Perturbations? NeurIPS

---

> ### Author Response · Authors · 2022-11-17
> **Paper revision v1.1**
>
> We have updated our paper to include results from an experiment that attacks a victim that has been patched to not pass prematurely (suggested by t1v2):
>
> 1. The resulting adversary is qualitatively very different from the first adversary we trained (see Figure 1 in the updated paper).
> 2. It wins over 98% of the time against a pass-patched victim with no search, and over 28% of the time against a pass-patched victim with enough search to be near-superhuman.
> 3. The resulting adversary still loses to a human amateur.
>
> t1v2's suggested experiment is actually still running and the latest results seem to suggest that an adversary trained for longer can win against victims which use 1000s of visits (well into superhuman territory). We hope to include these results in one final revision before the review period ends.

---

### Author Response · Authors · 2022-11-19
**Paper revision v1.2**

We have made one final update to our paper as this rebuttal period comes to a close. Compared to v1.1, v1.2 has a stronger adversary and also includes more analysis of how KataGo loses to our stronger adversary.

To summarise the major changes from our original submission (which our reviewers first read):

Via t1v2’s suggested experiment of attacking a pass-defended victim, we now have an improved adversary that can beat KataGo in a way that would be judged as legitimate in most human tournaments — addressing 6zA8’s concerns. Our improved adversary wins >99% of games against KataGo (no-search). Following t1v2's second suggestion, our adversary was trained against victims with search. This enabled the adversary to win >77% of games against KataGo playing with 2048 visits / move. At this level of search, KataGo is strongly superhuman. The improved adversary still loses to human amateurs.

---

### Decision · Program_Chairs · 2023-01-20

**Decision:**

Reject

**Justification For Why Not Higher Score:**

I would like to thank the authors for their effort to reply to the reviewers and update the manuscript to address their concerns, which improved the understanding the authors' work. Nevertheless, the overall novelty is limited and the contribution is not significant enough to warrant the acceptance of this work to ICLR.

**Justification For Why Not Lower Score:**

N/A

**Metareview: Summary, Strengths And Weaknesses:**

Summary:
The paper investigates attack against a state-of-the-art Go-playing AI system.

Strength:
Security concerns are of central interest in the machine learning community and investigating the adversarial robustness of modern reinforcement-learning-based systems is an interesting investigation.

Weakness:
Technical novelty is rather limited and the contribution is marginal.